# A Unified Federated Framework for Trajectory Data Preparation via LLMs

**Zhihao Zeng[1], Ziquan Fang[1], Wei Shao[1], Lu Chen[2], Yunjun Gao[2]**
[1]School of Software Technology, Zhejiang University
[2]College of Computer Science and Technology, Zhejiang University
`{zengzhihao,zqfang,wei.shao,luchen,gaoyj}@zju.edu.cn`

## Abstract

Trajectory data records the spatio-temporal movements of people and vehicles. However, raw trajectories are often noisy, incomplete, or inconsistent due to sensor errors and transmission failures. To ensure reliable downstream analytics, Trajectory Data Preparation (TDP) has emerged as a critical preprocessing stage, encompassing various tasks such as imputation, map matching, anomaly detection, trajectory recovery, compression, etc. However, existing TDP methods face two major limitations: (i) they assume centralized access to data, which is unrealistic under strict privacy regulations and data silo situations, and (ii) they train task-specific models that lack generalization across diverse or unseen TDP tasks. To this end, we propose FedTDP for Federated Trajectory Data Preparation (F-TDP), where trajectories are vertically partitioned across regions and cannot be directly shared. FedTDP introduces three innovations: (i) lightweight Trajectory Privacy AutoEncoder (TPA) with secret-sharing aggregation, providing formal privacy guarantees; (ii) Trajectory Knowledge Enhancer (TKE) that adapts LLMs to spatio-temporal patterns via trajectory-aware prompts, offsite-tuning, sparse-tuning, and bidirectional knowledge distillation; and (iii) Federated Parallel Optimization (FPO), which reduces communication overhead and accelerates federated training. We conduct experiments on 6 real-world datasets and 10 representative TDP tasks, showing that FedTDP surpasses 13 state-of-the-art baselines in accuracy, efficiency, and scalability, while also generalizing effectively across diverse TDP tasks.

## 1 Introduction

Trajectory data, which records the movement of people (Chen et al., 2024a) and vehicles (Miao et al., 2024b) in space and time, plays a critical role in traffic optimization and urban planning. However, raw trajectory data often contains inconsistencies (Liu et al., 2024b), noise (Fan et al., 2023), and missing values (Chib & Singh, 2024) due to sensor errors and communication failures. To improve data quality, **Trajectory Data Preparation (TDP)** has emerged as a crucial preprocessing stage, covering tasks such as data imputation (Musleh & Mokbel, 2023), map matching (Liu et al., 2024b), trajectory-user linking (Chen et al., 2024b), anomaly detection (Gao et al., 2023), trajectory recovery (Liu et al., 2024c), data compression (Fang et al., 2023), etc. These tasks are indispensable for enabling robust downstream analytics, such as traffic forecasting, trajectory clustering, and anomaly explanation. Note that, each of these TDP tasks has attracted considerable attention in the spatio-temporal community (Liu et al., 2024b; Miao et al., 2024a; Hu et al., 2024a). Despite their considerable efforts, existing TDP methods have **two limitations** in the following.

**L1: Decentralized requirement.** Trajectory data is highly sensitive, as it encodes individuals' movements. To safeguard such data, strict privacy regulations have been enacted worldwide. For example, the Federal Geographic Data Committee (FGDC, 2015) and China's Personal Information Protection Law (PIPL, 2021) explicitly restrict the sharing of raw mobility data across areas. In practice, platforms (e.g., Uber Movement) split trips along administrative boundaries to comply with privacy rules (UMP, 2022). This leads to **vertical partitioning**, where each region only observes partial trajectories. Models trained only on partial trajectories suffer from boundary discontinuities and biased spatio-temporal patterns, which severely impact TDP tasks such as imputation or anomaly detection near administrative borders (Musleh & Mokbel, 2023; Fang et al., 2023). Fig. 1 shows an

example using the GeoLife dataset (Zheng et al., 2010), where trajectories crossing administrative regions (depicted in different colors) are split into disjoint sub-trajectories, each managed by a local data silo. *However, existing federated learning studies mostly focus on horizontal partitioning, where different users' data reside at different institutions. In contrast, the vertical partitioning of trajectories across regions has not been systematically studied.*

**L2: Lack of a general TDP framework.** Existing TDP methods are typically designed for a *single, narrowly defined task*, which makes them hard to generalize. For example, map-matching methods (e.g., HMM-based alignment or deep neural map-matching) are effective only for projecting GPS points onto road networks; trajectory imputation models (e.g., RNN- or GAN-based sequence completion) are restricted to handling missing points; and trajectory denoising or anomaly detection algorithms often rely on handcrafted spatio-temporal features tailored to specific datasets. Each of these TDP methods must be retrained or redesigned when applied to a different or unseen TDP task. *As a result, the overall pipeline remains fragmented, with high computational cost and limited scalability. Moreover, the diversity of TDP tasks makes it infeasible to adapt single-task models; thus, a unified approach is required.*

**These two limitations motivate us to study a new problem, i.e., Federated Trajectory Data Preparation (F-TDP).** Given vertically partitioned trajectories stored in different regional silos, we aim to collaboratively perform multiple TDP tasks without sharing raw data across silos.

**Design Motivation.** To address privacy concerns in this setting, we build on **Federated Learning (FL)**, a distributed paradigm that enables clients to train models collaboratively while keeping data local (Liu et al., 2024a; Miao et al., 2025). Specifically, we treat each region as a client that stores partial trajectories and

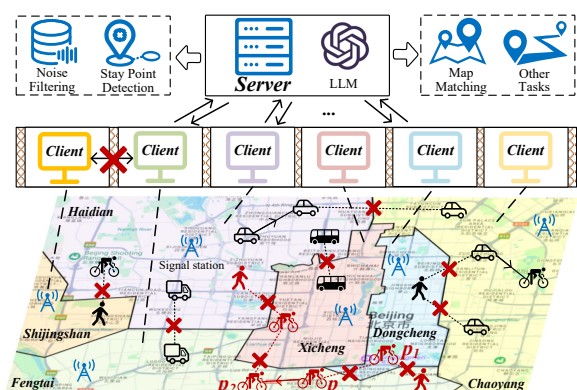

Figure 1: Federated trajectory data preparation

contributes to a federated server for model training, thus preserving privacy in F-TDP. As shown in Fig. 1, FL enables multiple **Clients** (i.e., regions) to collaboratively train a model on a **Server** while keeping trajectory data decentralized, thereby preserving the data privacy of each client. Meanwhile, **Large Language Models (LLMs)** have demonstrated strong multi-task generalization in diverse domains, including mobility applications (Fang et al., 2026) and trajectory analysis (Du et al., 2024). This suggests the potential of developing a **TDP-oriented LLM** to unify different TDP tasks within a unified framework, avoiding repeated training of task-specific models across silos.

However, **directly combining FL and LLMs** to perform TDP tasks is insufficient, as F-TDP raises **three unique challenges** that existing methods fail to address.

**C1: Privacy.** TDP tasks often necessitate considering the data context Gao et al. (2023); Musleh & Mokbel (2023); Liu et al. (2024b), which involves the exchange and sharing of data and demands collaborative processing across clients (i.e., cross-client TDP), raising privacy concerns. Such data sharing may reveal highly sensitive information, including: home/work locations, fine-grained movement patterns, lifestyle routines, and cross-border movements of individuals. As shown in Fig. 1, if the data p is missing, the Fengtai region needs to utilize the context of the missing data (i.e., $p$, and $p_2$) for data imputation. However, due to data privacy constraints, the Fengtai region cannot access $p_1$ from the Dongcheng region. Consequently, ensuring the privacy of trajectory data thus constitutes the challenge the FedTDP framework must address.

**C2: Trajectory Knowledge Learning.** Existing LLMs are primarily designed for textual corpora (Fan et al., 2024; Lee et al., 2022), and thus fail to capture the unique spatio-temporal properties of trajectories (Musleh & Mokbel, 2023; Fang et al., 2023), such as temporal regularity and spatial dependency. Moreover, their pre-training relies on large-scale unsupervised text corpora (Eltabakh et al., 2024; Gu et al., 2023), which lack the intricate mobility patterns required by TDP tasks (Liu et al., 2024b; Hu et al., 2024b). As a result, directly applying LLMs to trajectory data preparation leads to suboptimal performance. A key challenge is therefore to develop a *TDP-oriented LLM* that can effectively bridge this gap and generalize across diverse preparation tasks.

**C3: Efficiency.** Deploying and training LLMs locally on clients is infeasible due to limited computational resources and storage capacities (Sun et al., 2024; Zhang et al., 2024a; Hou et al., 2024). As a result, LLMs are often hosted on servers, which introduces both communication overhead and storage burdens (Liu et al., 2025), while wasting computational resources on the client side. Furthermore, LLMs contain hundreds of millions of parameters, and even parameter-efficient fine-tuning (PEFT) techniques (Hu et al., 2022) remain highly time- and resource-intensive under federated settings, as proved by **Appendix 5.4**. The challenge is thus to design *scalable optimization strategies* that reduce communication cost and accelerate training while maintaining high accuracy.

**Contributions.** To the best of our knowledge, no existing studies (**Related Work** in **Appendix A**) address the above challenges. In this paper, we propose **FedTDP**, a unified federated framework for trajectory data preparation via LLMs. Our main contributions are as follows:

- **Problem formulation of F-TDP.** We formally define **Federated Trajectory Data Preparation (F-TDP)** and provide a unified framework addressing vertical partitioning and multi-task challenges, which to our knowledge, has not been systematically studied before.
- **Trajectory-specific privacy mechanism.** We design a lightweight **Trajectory Privacy AutoEncoder (TPA)**. Unlike generic embedding encoders, TPA is explicitly designed to preserve spatio-temporal continuity while protecting privacy. To resist embedding/gradient inversion, TPA integrates a decentralized secret-sharing aggregation protocol, with **formal privacy guarantees** provided in **Appendix C.2**.
- **TDP-oriented LLM with trajectory knowledge enhancement.** We develop a **Trajectory Knowledge Enhancer (TKE)** to bridge the gap between text-oriented LLMs and trajectory data. TKE incorporates trajectory-aware prompt engineering, offsite-tuning, and bidirectional knowledge distillation between server-side LLMs and client-side SLMs, enabling effective cross-task generalization for 10 TDP tasks.
- **Federated parallel optimization for efficiency.** We propose **Federated Parallel Optimization (FPO)**, which coordinates the training of TPA, SLMs, and LLMs under vertical FL constraints. By combining split learning, alternating optimization, and parallel execution, FPO reduces communication overhead and significantly accelerates convergence.

## 2 PRELIMINARY

The frequently used **notations and descriptions** in this paper are shown in **Appendix B**.

**Definition 1 (Spatio-Temporal Point).** *A spatio-temporal point is represented as $p = \langle l, t \rangle$, where $l = (lon, lat)$ is a tuple of longitude and latitude location coordinates, and $t$ refers to the observed time associated with this spatio-temporal point.*

**Definition 2 (Trajectory).** *A trajectory comprises chronological spatio-temporal points, denoted as $T = \{p_1, p_2, \ldots\}$, which typically represents the movement of a user. In addition, a trajectory can be segmented into multiple sub-trajectories, denoted as $T = \{ST^{(1)}, ST^{(2)}, \ldots\}$.*

**Definition 3 (Data Silo).** *A data silo $S$ has its own collected trajectory dataset $D$. In federated learning, a data silo $S$ is represented as a client $C$, typically a regional data storage platform or institution, responsible for the collection and management of trajectory data within that region. Specifically, a trajectory $T = \{p_1, p_2, \ldots\}$ is segmented into sub-trajectories based on the geographic locations, denoted as $T = \{ST^{(C_1)}, ST^{(C_2)}, \ldots\}$, where sub-trajectory $ST^{(C_i)}$ is stored in client $C_i$.*

**Problem Formulation (F-TDP).** Given the server's LLM $\theta_{LLM}$ and the trajectory dataset $\mathcal{D} = \{D_1, D_2, \ldots\} \rightarrow \{T_1, T_2, \ldots\}$ of all clients $\mathcal{C} = \{C_1, C_2, \ldots\}$, where client $C_i$ holds dataset $D_i$ and $T_i$ is the $i$-th complete trajectory, F-TDP is to employ $\theta_{LLM}$ on $\mathcal{D}$ for performing various trajectory data preparation tasks, where collected trajectories $D_i$ of client $C_i$ cannot be shared and exchanged to the server and other clients, i.e., *F-TDP*$(\mathcal{D}) = \theta_{LLM}(T_i)$ where $T_i = \{ST_i^{(C_1)}, ST_i^{(C_2)}, \ldots\}$ and $\theta_{LLM}(T_i)$ is the result of $\theta_{LLM}$ on the trajectory $T_i$, with different forms of output depending on the TDP task, such as the cleaned trajectory, points, or classification result.

## 3 TRAJECTORY DATA PREPARATION TASK

We demonstrate all major types of TDP tasks, with the **illustration examples** shown in **Appendix B**.

Figure 2: The overview of our framework

**T-1: Anomaly Detection (*AD*).** It aims to detect trajectories that deviate significantly from typical movement patterns. These anomalies may arise from unusual user behavior or data collection errors.

**T-2: Trajectory Imputation (*TI*).** It aims to reconstruct a complete trajectory by estimating the missing points based on available spatio-temporal points. This often occurs when GPS signals are lost or data collection is interrupted.

**T-3: Noise Filtering (*NF*).** It aims to identify and remove irrelevant spatio-temporal points that deviate from the trajectory. These noisy points often arise from GPS inaccuracies or sensor malfunctions.

**T-4: Stay Point Detection (*SPD*).** It aims to identify locations where a moving object stays in an area for a certain duration. A stay point usually represents a point of interest, such as residence, office.

**T-5: Map Matching (*MM*).** It aims to map the spatio-temporal point to the most probable segment in the road network. This is often the case when there is a deviation in the collected GPS position.

**T-6: Trajectory-User Link (*TUL*).** It aims to link an anonymous trajectory with its corresponding user. These trajectories are often collected without any user-identifying information.

**T-7: Travel Mode Identification (*TMI*).** It aims to identify the travel mode based on the moving pattern of trajectory, which is walking, biking, taking the bus, or driving a car.

**T-8: Trajectory Simplification (*TSim*).** It aims to reduce the number of spatio-temporal points in a trajectory while preserving its essential shape and features.

**T-9: Trajectory Segmentation (*TSeg*).** It aims to divide a trajectory into meaningful segments based on specific criteria such as stay points or travel modes.

**T-10: Trajectory Recovery (*TR*).** It aims to reconstruct a complete trajectory from partially observed spatio-temporal points. This often occurs when some parts of the trajectory are missing or unobserved.

## 4 OUR APPROACH

Fig. 2 provides an overview of the **FedTDP** framework, which involves a central server and multiple regional clients. FedTDP consists of three core modules—the *Trajectory Privacy AutoEncoder (TPA)*, the *Trajectory Knowledge Enhancer (TKE)*, and the *Federated Parallel Optimization (FPO)*—each designed to address one of the key challenges in the F-TDP problem.

To enable distributed computation, we introduce the notion of a **Small Language Model (SLM)**, a lightweight counterpart of the server-side LLM, which is deployed on each client. The SLM is used whenever a client can process its local sub-trajectory $ST^{(C)}$ independently, thereby reducing server workload and leveraging local computational resources. If $ST^{(C)}$ requires cross-client collaboration (e.g., when trajectories span multiple regions), the encoded representations are transmitted to the server, where the LLM is used for global TDP. The overall workflow is as follows.

- **Local TDP.** For local tasks, TKE generates a trajectory-specific prompt and feeds it to the client's SLM (①–②). The SLM produces preliminary results, which are further refined by TKE (③–④).

- **Cross-client TDP.** For tasks requiring collaboration across silos, TPA first encodes local trajectories into secure embeddings before transmission (①–②). FPO then freezes the transmitted data to reduce communication overhead (③). On the server side, TKE generates prompts for the LLM (④–⑤), and TPA subsequently decodes the LLM outputs back into trajectory representations (⑥–⑦). Finally, FPO manages the return flow (⑧), and TKE refines the results at the client level (⑨–④).

### 4.1 TRAJECTORY PRIVACY AUTOENCODER (TPA)

**Design Motivation**. As aforementioned, F-TDP involves the joint processing of data from multiple clients, i.e., cross-client TDP, necessitating data exchange and sharing. Consequently, safeguarding the privacy of trajectory data becomes essential. Although differential privacy (Dwork et al., 2006)

can be applied to ensure data privacy, it requires adding noise to the data, which diminishes its utility and reduces model accuracy. In contrast, FedTDP proposes a lightweight Trajectory Privacy AutoEncoder (TPA) to protect trajectory data privacy while maintaining spatio-temporal correlations.

Specifically, the TPA module employs an encoder-decoder architecture that encodes trajectory data $T = \{p_1, p_2, \ldots\}$ into embeddings $E = \{e_1, e_2, \ldots\}$, where each spatio-temporal point $p_i$ is independently encoded as $e_i = \theta_{Enc}(p_i)$. Then, these clients' embeddings are transmitted to the server for aggregation and alignment via the anonymized user identifier, i.e., $\mathcal{E} = \bigcup_{i=1}^{|\mathcal{C}|} E_i$, preserving both intra- and inter-client spatio-temporal dependencies (e.g., speed, direction), which helps the LLM to capture spatio-temporal relationships in the trajectory data. Next, the server splits and distributes results $\tilde{\mathcal{E}} = \{\tilde{e_1}, \tilde{e_2}, \ldots\}$ outputted by the LLM to clients, where the decoder reconstructs the estimated trajectory $\tilde{T} = \{\tilde{p_1}, \tilde{p_2}, \ldots\}$ through $\tilde{p_i} = Dec(\tilde{e_i})$. Here, TPA is implemented as a lightweight three-layer MLP (Rosenblatt, 1958) with GELU (Hendrycks & Gimpel, 2016) activation, 256 hidden dimensions, and 32 embedding dimensions, which does not introduce significant computational overhead, as also proved in the ablation study (see Section 5.2).

However, merely using embeddings for transmission cannot safeguard data privacy completely in FL, as attackers can recover the raw data by embedding (Chen et al., 2024c; Huang et al., 2024) and gradient (Zhu et al., 2019; 2023) inversion attacks during TPA model aggregation. Specifically, traditional FL model aggregation, which exchanges client gradients and aggregated parameters, are vulnerable to these attacks. While homomorphic encryption (Rivest et al., 1978) and differential privacy offer solutions, they introduce computational overhead or degrade model accuracy. In contrast, we propose a decentralized aggregation approach based on secret sharing (Shamir, 1979), achieving secure TPA aggregation without compromising training efficiency or model accuracy.

Initially, each client pair $(C_i, C_j)$ generates a shared secret key $sk_{i,j} = sk_{j,i}$ stored locally, respectively. Then, the TPA parameters are partitioned into $|\mathcal{C}|$ parameter blocks $\{P^{(1)}, P^{(2)}, \ldots\}$. For aggregation, the client $C_i$ masks its parameter block using secret keys $\{sk_{i,0}, sk_{i,1}, \ldots\}$ determined with the other clients to mask parameter blocks, adding $sk_{i,j}$ if $i > j$ or subtracting it if $i < j$, as shown below:

$$\tilde{P}_i^{(k)} = P_i^{(k)} + \sum_{j=1 \& j \neq i}^{|\mathcal{C}|} a_{i,j} * sk_{i,j}, \quad a_{i,j} = \left\{ \begin{array}{ll} 1, & i < j \\ -1, & i > j \end{array} \right. , \tag{1}$$

where client $C_i$ holds the parameter block $P_i^{(k)}$ and $\tilde{P}_i^{(k)}$ is the mask parameter block.

**Theorem 1.** *Given the mask parameter blocks $\{\tilde{P}_1^{(k)}, \tilde{P}_2^{(k)}, \ldots\}$ from all clients, the result of aggregating them is equal to the result of aggregating raw parameter blocks $\{P_1^{(k)}, P_2^{(k)}, \ldots\}$ for all clients:*

$$\sum_{i=1}^{|\mathcal{C}|} \tilde{P}_i^{(k)} = \sum_{i=1}^{|\mathcal{C}|} P_i^{(k)} \tag{2}$$

*Proof.* The detailed **proofs of Theorem 1** are provided in **Appendix C.1**. □

According to Theorem 1, the client $C_k$ can obtain the aggregation result $\overline{P}^{(k)}$ of the parameter block $P^{(k)}$ by aggregating the mask parameter blocks transmitted from clients, as formally shown below:

$$\overline{P}^{(k)} = \frac{1}{|\mathcal{C}|} \sum_{i=1}^{|\mathcal{C}|} \tilde{P}_i^{(k)} = \frac{1}{|\mathcal{C}|} \sum_{i=1}^{|\mathcal{C}|} P_i^{(k)} \tag{3}$$

Finally, the aggregated parameter block is broadcast to clients for the TPA model updates.

**Approach Analysis.** Note that TPA is fundamentally different from differential privacy (DP), not only in whether noise is added but also in its underlying privacy paradigm. DP achieves privacy by injecting random noise into data, which inevitably distorts spatio-temporal correlations that are essential for TDP tasks. In contrast, TPA is a deterministic, learning-based transformation that removes privacy-sensitive details while preserving useful patterns. Unlike DP, which requires careful calibration of noise scales and suffers from a utility–privacy tradeoff, TPA avoids this tension by design. Moreover, while DP-perturbed trajectories can still leak probabilistic information about the original locations or time ranges within a privacy budget (Yao et al., 2022; Zhang et al., 2024b), TPA provides a *different but strong* guarantee: as long as the encoder and decoder remain private, reconstructing the raw trajectory is computationally infeasible (see **Appendix C.2**).

## 4.2 TRAJECTORY KNOWLEDGE ENHANCER (TKE)

**Design Motivation**. Since existing LLMs are designed for text data and contain only general textual knowledge (Fan et al., 2024; Lee et al., 2022; Eltabakh et al., 2024), they cannot be directly applied to trajectory data and TDP tasks. Although a few spatio-temporal LLMs (Li et al., 2024c; Zhang et al., 2023b; Li et al., 2023) have been proposed, none of them have considered TDP. In contrast, to develop a TDP-oriented LLM, FedTDP designs Trajectory Knowledge Enhancer (TKE) that consists of trajectory prompt engineering, trajectory offsite-tuning, LoRA sparse-tuning, and bidirectional knowledge learning, to enhance the model learning abilities on TDP knowledge.

**i) Trajectory Prompt Engineering** To help the SLM and LLM understand trajectory data and learn TDP knowledge, TKE designs a trajectory instruction paradigm to generate the TDP prompt, defined as (***Task***, ***Data***, ***Information***, ***Format***). Specifically, ***Task*** is the textual instruction consisting of the task name and the task description, as listed in Section 3. ***Data*** is the input trajectory data, either as trajectory data $T = \{p_1, p_2, \ldots\}$ to the SLM for local TDP or embeddings $E = \{e_1, e_2, \ldots\}$ to the LLM for cross-client TDP. ***Information*** is the optional trajectory context (e.g., road network, weather) from public sources such as OpenStreetMap (2025) and OpenWeatherMap (2025), to enhance the model's ability to perform TDP tasks, which is manually configured and task-specific rather than determined automatically by the model. ***Format*** is a task-specific output format, such as classification results for TDP tasks including AD, TUL, and TMI, trajectories for TDP tasks including TI, NF, TSim, TSeg, MM, and TR, and spatio-temporal points for the SPD task. A few representative TDP tasks constructed via **trajectory prompt engineering** are provided in **Appendix C.3** to illustrate the flexibility and applicability of our framework.

**ii) Trajectory Offsite-Tuning.** To enhance the learning capabilities of the SLM in clients, TKE employs the LLM to assist it in learning trajectory knowledge by trajectory off-site tuning. Specifically, inspired by the offsite-tuning (Xiao et al., 2023), we divide the LLM into two components, denoted as $\theta_{LLM} = [\mathcal{A}, \mathcal{F}]$. Here, the adapter $\mathcal{A}$ is the last few layers of the LLM to specialize general features for specific tasks, enabling task-specific feature mapping and decision making. Besides, the foundation $\mathcal{F}$ is the remaining layers excluding $\mathcal{A}$, to extract general data features, transforming raw inputs into meaningful representations. Initially, it dispatches the server's adapter $\mathcal{A}$ to the client as the final few layers to be integrated into the client's SLM. Consequently, the SLM is composed of two components, denoted as $\theta_{SLM} = [\mathcal{A}, \mathcal{F}']$, where $\mathcal{F}'$ is the foundation of the SLM. Subsequently, the SLM employs LoRA to reduce the number of parameters in the adapter and then transmits the fine-tuning adapter to the server for aggregation and updates.

Note that FedTDP does not simply transfer the trained LLM adapter to the SLM. Instead, the adapter $\mathcal{A}$ is leveraged to *augment the SLM's learning capacity* during training. Specifically, during tuning, $\mathcal{A}$ is dispatched to the client and attached to the tail of the local SLM. The client fine-tunes only $\mathcal{A}$ on its local data, and the updated adapters are then returned to the server for aggregation. To ensure compatibility across different model families, the only requirement is *hidden dimension alignment*: the output dimension of the SLM's final layer must match the input dimension of the adapter $\mathcal{A}$.

**iii) LoRA Sparse-Tuning.** To reduce the number of training parameters, TKE proposes LoRA sparse-tuning, as shown in Fig. 3. According to works on sparsity (Alistarh et al., 2018; Dai et al., 2022; Zhang et al., 2023a), more significantly varying parameters have a greater contribution to model convergence. Therefore, we only choose the layer in the SLM where the LoRA parameter change rate is the top $m$ for training. Specifically, the client calculates the ratio of the LoRA parameters change rate of each layer to the global LoRA parameters change rate of all $N$ layers ("ratio" for short), as shown below:

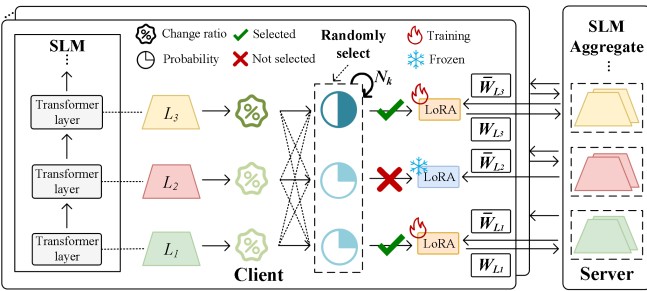

Figure 3: LoRA sparse-tuning

$$R^{(r)}(L_i) = \frac{CR^{(r)}(L_i)}{\sum_{j=1}^{N} CR^{(r)}(L_j)} , \qquad (4)$$

where $CR^{(r)}(L_i)$ is the LoRA parameters change rate of layer $L_i$ at round $r$, as shown below:

$$CR^{(r)}(L_i) = |\frac{L_i^{(r)} - L_i^{(r-1)}}{L_i^{(r-1)}}| \tag{5}$$

Then, we randomly select $M = \lfloor m * N \rfloor$ layers to participate in the next round of the SLM training.

**Theorem 2.** *Given the ratio $R^{(r)}(L_i)$ of layer $L_i$ and the number of layers $M$ to be trained, the probability $P(L_i, M)$ of layer $L_i$ to train is shown:*

$$
\begin{aligned}
P(L_i, M) = R(L_i) + R(L_i) \sum_{\substack{j_1 \\ j_1 \neq i}}^{N} \frac{R(L_{j_1})}{1 - R(L_{j_1})} + \cdots \\
+ R(L_i) \sum_{\substack{j_1, \ldots, j_{M-1} \\ j_t \neq i, \ j_a \neq j_b}}^{N} \frac{R(L_{j_1}) \cdots R(L_{j_{M-1}})}{(1 - R(L_{j_1})) \cdots (1 - \sum_{t=1}^{M-1} R(L_{j_t}))}
\end{aligned}
\tag{6}
$$

*Proof.* The detailed **proofs of Theorem 2** are provided in **Appendix C.4**. □

According to Theorem 2, it chooses the training layers based on their probability at each training round. Finally, the client uploads the LoRA parameters of the trained layers to the server for aggregation, and the server assigns different weights to the parameters based on the number of clients involved in training on these layers, as formally shown below:

$$\overline{W}_{L_i}^{(r)} = \frac{(|\mathcal{C}| - |\mathcal{C}'|) * \frac{\sum_{j=1}^{|\mathcal{C}'|} n_j * W_{L_i,j}^{(r)}}{\sum_{j=1}^{|\mathcal{C}'|} n_j} + \overline{W}_{L_i}^{(r-1)}}{|\mathcal{C}| - |\mathcal{C}'| + 1}, \tag{7}$$

where $W_{L_i,j}^{(r)}$ is the LoRA parameters of layer $L_i$ sent by client $C_j$ at training round $r$, $\overline{W}_{L_i}^{(r)}$ is the aggregated LoRA parameters, and $|\mathcal{C}'|$ is the number of clients that have trained layer $L_i$.

**iv) Bidirectional Knowledge Learning.** To improve the model learning capabilities, TKE develops bidirectional knowledge learning to enhance their TDP knowledge. Specifically, in order for the SLM to learn useful TDP knowledge in the complex output space of the LLM, it aligns the SLM's output with LLM's high frequency output using the inverse KL divergence (Kullback & Leibler, 1951):

$$\min_{\theta_{SLM}} D_{KL}(P_{\theta_{SLM}} || P_{\theta_{LLM}}) = \sum_T P_{\theta_{SLM}}(T) \log(\frac{P_{\theta_{SLM}}(T)}{P_{\theta_{LLM}}(T)}), \tag{8}$$

where $P_{\theta_{SLM}}$ and $P_{\theta_{LLM}}$ are the output distribution of the SLM and LLM, respectively. Besides, since the SLM can access raw trajectory data, it aligns the LLM's output with the SLM's output using the forward KL divergence, which enables the LLM to learn the trajectory knowledge of the SLM:

$$\min_{\theta_{LLM}} D_{KL}(P_{\theta_{SLM}} || P_{\theta_{LLM}}) = \sum_T P_{\theta_{SLM}}(T) \log(\frac{P_{\theta_{SLM}}(T)}{P_{\theta_{LLM}}(T)}) \tag{9}$$

### 4.3 FEDERATED PARALLEL OPTIMIZATION (FPO)

**Design Motivation**. To improve training efficiency, FedTDP introduces Federated Parallel Optimization (FPO), which utilizes split learning, alternating optimization, and parallel training to reduce data transmission and enhance the training parallelism. The overall process of the FPO module is shown in Fig. 4.

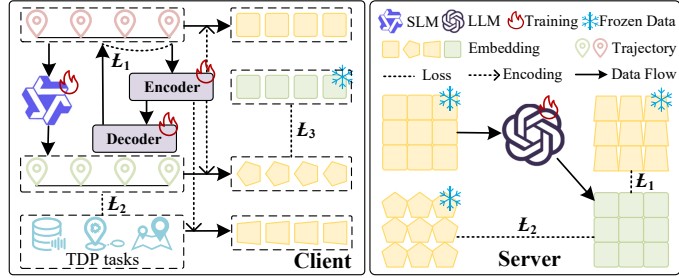

Figure 4: Federated parallel optimization

First, to enable the simultaneous training of the client and server, it employs split learning to decompose the federated training process into client and server training. Specifically, the client is responsible for the training of the TPA model (i.e., the encoder and decoder) and SLM, while the server manages the training of the LLM. Besides, to reduce data transmission, it utilizes alternating optimization to freeze the data required by the client and server, respectively. During training, the server freezes the embeddings uploaded by the

Table 1: The evaluated trajectory data preparation (TDP) tasks

| Type | Category | Task | Dataset |
|------|----------|------|---------|
| **Seen** (seen in training) | Data Cleaning | Anomaly Detection (**AD**) 
 Trajectory Imputation (**TI**) | Geolife |
| | Data Matching | Map Matching (**MM**) 
 Trajectory User Linking (**TUL**) | |
| | Data Annotation | Travel Mode Identification (**TMI**) | |
| | Data Reduction | Trajectory Simplification (**TSim**) | |
| | Data Augmentation | Trajectory Recovery (**TR**) | |
| **Unseen** (unseen in training) | Data Cleaning | Anomaly Detection 
 Trajectory Imputation | Porto |
| | | Noise Filtering (**NF**) 
 Stay Point Detection (**SPD**) | T-Drive |
| | Data Matching | Map Matching | Tencent |
| | | Trajectory User Linking | Gowalla |
| | Data Annotation | Travel Mode Identification | SHL |
| | Data Reduction | Trajectory Segmentation (**TSeg**) | |
| | | Trajectory Simplification | T-Drive |
| | Data Augmentation | Trajectory Recovery | |

client for the LLM training, while the client freezes the results outputted by the server's LLM for the TPA model and SLM training. Finally, to enhance the training parallelism, it uses parallel training to optimize several objectives in parallel. Specifically, the client has three optimization objectives: (i) minimizing the reconstruction loss $\mathcal{L}_1$ of the TPA model, (ii) reducing the inverse KL loss $\mathcal{L}_2$ between SLM and LLM outputs, and (iii) minimizing the loss $\mathcal{L}_3$ between SLM outputs and labels. On the other hand, the server has two optimization objectives: (i) minimizing the forward KL loss $\mathcal{L}_1$ between LLM and SLM outputs, and (ii) reducing the loss $\mathcal{L}_2$ between LLM outputs and labels.

The **multi-task training** and a few qualitative **examples of LLM/SLM outputs** on different TDP tasks are shown in **Appendix C.5** and **Appendix C.6**, respectively.

## 5 EXPERIMENT

**Datasets.** We evaluate the framework on the 10 mainstream TDP tasks in Table 1, which are widely studied in the communities (Liu et al., 2024b; Musleh & Mokbel, 2023; Hu et al., 2024a). For seen tasks, we train FedTDP via a unified multi-task learning objective using 10% of **GeoLife** (Zheng et al., 2010) dataset. It contains various quality issues such as positional inaccuracies, data noise, and lower precision, which makes it suitable for various tasks. For the unseen tasks, the following datasets are used: (i) **Porto (2015)**, (ii) **T-Drive** (Yuan et al., 2010), (iii) **Tencent** (Liu et al., 2024b), (iv) **Gowalla** (Cho et al., 2011), and (v) **SHL (2017)**. Dataset details are provided in **Appendix D.1**.

**Baselines.** As FedTDP is the first framework for LLM-based general trajectory data preparation, we compare FedTDP against three categories of representative baselines: **(i) Single-Task methods** (referred to as **S-TDP**). We include ATROM (Gao et al., 2023) (anomaly detection), Kamel (Musleh & Mokbel, 2023) (trajectory imputation), GraphMM (Liu et al., 2024b) (map matching), AttnTUL (Chen et al., 2024b) (trajectory-user link), Estimator (Hu et al., 2024a) (travel mode identification), S3 (Fang et al., 2023) (trajectory simplification), and LightTR (Liu et al., 2024c) (trajectory recovery). These methods are recognized as leading task-specific approaches in their respective domains and serve as strong single-task baselines. **(ii) LLM-based general data preparation methods.** Since FedTDP is the first proposal for general trajectory data preparation settings, we compare FedTDP with three state-of-the-art approaches that leverage LLMs for general table data preparation, namely FM4DP (Narayan et al., 2022), MELD (Yan et al., 2024), and TableGPT (Li et al., 2024a). **(iii) LLM-based spatio-temporal data analysis methods.** Finally, we consider three recent state-of-the-art methods that employ LLMs for spatio-temporal modeling, including PromptGAT (Da et al., 2024), UniST (Yuan et al., 2024), and UrbanGPT (Li et al., 2024b). For LLM-based baselines, they directly input raw trajectory data into LLMs to perform TDP tasks.

To ensure fairness, all LLM-based baselines are applied under identical data preprocessing as FedTDP, ensuring that performance differences primarily reflect the contribution of our framework rather than implementation bias. More **implementation details** and **evaluation metrics** are provided in **Appendix D.2** and **Appendix D.3**, respectively.

### 5.1 OVERALL PERFORMANCE

Table 2 reports the overall performance of FedTDP compared with all baselines across different datasets and tasks, under both few-shot and zero-shot scenarios ("–" denotes unsupported tasks). First, FedTDP consistently outperforms single-task TDP methods, achieving at least **18.38%** improvement.

Table 2: The overall performance on various trajectory data preparation tasks (few-shot/zero-shot)

| Dataset | Task | Metric[1] | S-TDP[2] | FM4DP | MELD | TableGPT | PromptGAT | UniST | UrbanGPT | FedTDP |
|---|---|---|---|---|---|---|---|---|---|---|
| GeoLife (Seen) | AD | $F_1$ | 60.36/57.75 | 52.09/47.84 | 54.52/48.21 | 51.58/49.44 | 66.24/66.12 | 70.26/70.14 | 74.39/70.31 | **81.20/80.04** |
| | | Acc | 73.87/71.24 | 64.50/53.43 | 65.48/54.80 | 56.03/53.71 | 73.18/70.42 | 77.85/74.65 | 79.38/78.23 | **87.45/87.38** |
| | TI | $F_1$ | 69.48/65.92 | 62.12/56.10 | 62.67/58.29 | 61.15/56.77 | 73.14/70.73 | 76.28/71.67 | 76.86/70.73 | **82.89/81.26** |
| | | Acc | 78.28/67.61 | 69.79/60.59 | 69.45/67.59 | 70.98/61.32 | 75.15/72.90 | 80.81/77.21 | 81.29/73.78 | **94.99/92.07** |
| | MM | $F_1$ | 39.75/36.22 | 28.96/25.75 | 34.59/34.58 | 30.11/26.45 | 47.44/44.74 | 53.11/47.47 | 56.91/54.51 | **65.32/64.85** |
| | | Acc | 40.05/39.37 | 29.58/28.45 | 33.75/38.80 | 33.08/35.67 | 48.19/46.14 | 48.44/42.82 | 53.17/51.13 | **76.48/74.25** |
| | TUL | $F_1$ | 22.06/21.27 | 18.01/17.15 | 17.75/15.08 | 17.35/15.43 | 36.72/34.55 | 38.42/37.47 | 46.56/45.17 | **54.79/54.44** |
| | | Acc | 29.85/28.78 | 23.66/21.02 | 23.29/21.79 | 24.00/23.79 | 37.88/35.31 | 41.40/40.53 | 49.86/46.69 | **64.84/58.55** |
| | TMI | $F_1$ | 69.05/62.82 | 57.29/55.76 | 59.03/55.84 | 57.01/51.94 | 66.98/62.26 | 69.29/68.45 | 69.45/66.20 | **82.69/80.48** |
| | | Acc | 79.98/77.82 | 69.20/64.28 | 70.56/64.07 | 66.14/65.38 | 70.06/64.85 | 76.72/68.57 | 75.13/67.77 | **89.66/83.16** |
| | TSim | SED | 1.88/1.95 | 2.29/2.53 | 1.89/2.01 | 1.96/2.29 | 1.83/1.95 | 1.57/1.71 | 1.69/1.89 | **1.39/1.55** |
| | | DAD | 0.59/0.70 | 0.82/1.08 | 0.87/0.92 | 0.81/0.96 | 0.80/0.89 | 0.69/0.80 | 0.55/0.71 | **0.43/0.53** |
| | TR | $F_1$ | 57.16/53.16 | 41.69/39.36 | 41.46/40.55 | 44.46/42.22 | 58.45/56.39 | 60.43/56.73 | 62.97/57.53 | **66.89/65.54** |
| | | Acc | 59.00/55.89 | 44.48/42.90 | 53.73/45.57 | 48.20/45.41 | 60.40/58.99 | 64.23/57.23 | 62.72/62.51 | **70.03/67.12** |
| Porto | AD | $F_1$ | 35.57/33.03 | 46.99/40.19 | 47.76/36.59 | 44.65/42.73 | 48.12/45.84 | 53.54/46.26 | 51.93/47.94 | **68.78/62.91** |
| | | Acc | 39.59/37.98 | 46.84/43.65 | 49.47/41.12 | 53.45/47.58 | 53.89/49.57 | 58.89/50.61 | 55.98/50.39 | **69.59/65.46** |
| | TI | $F_1$ | 17.36/5.59 | 35.23/23.05 | 40.14/27.80 | 36.04/26.12 | 42.64/40.49 | 42.77/48.35 | 43.13/42.19 | **62.44/56.91** |
| | | Acc | 19.57/9.73 | 39.47/29.65 | 47.77/31.72 | 45.62/29.56 | 46.54/42.35 | 48.88/52.76 | 48.94/45.70 | **63.90/57.33** |
| T-Drive | NF | $F_1$ | – | 20.91/14.93 | 27.42/22.76 | 24.37/13.32 | 49.00/40.64 | 44.87/44.15 | 54.46/45.07 | **61.90/56.18** |
| | | Acc | | 25.40/17.95 | 34.46/23.42 | 32.26/15.35 | 55.97/43.70 | 46.49/45.06 | 57.11/49.48 | **64.65/56.33** |
| | SPD | $F_1$ | – | 28.49/16.85 | 24.24/13.90 | 26.72/20.08 | 50.77/47.11 | 51.89/48.44 | 57.28/47.01 | **63.84/56.04** |
| | | Acc | | 31.76/20.09 | 38.52/21.04 | 39.70/22.35 | 53.09/51.81 | 58.60/49.95 | 55.84/48.96 | **63.65/59.44** |
| | TSim | SED | 3.10/3.40 | 3.77/3.51 | 3.44/3.25 | 3.23/3.32 | 3.20/3.42 | 3.40/3.45 | 3.19/3.20 | **2.11/2.83** |
| | | DAD | 0.82/0.90 | 0.68/0.82 | 0.91/0.98 | 0.61/0.81 | 0.77/0.84 | 0.77/0.81 | 0.75/0.84 | **0.56/0.72** |
| | TR | $F_1$ | 19.34/2.65 | 27.99/13.60 | 34.81/19.06 | 24.91/16.62 | 38.57/36.17 | 43.88/39.32 | 46.71/41.10 | **48.97/48.85** |
| | | Acc | 21.22/5.82 | 31.69/26.71 | 35.73/21.77 | 35.50/24.14 | 42.05/41.05 | 46.14/43.78 | 48.14/43.39 | **52.44/49.83** |
| Tencent | MM | $F_1$ | 15.39/14.40 | 30.16/18.71 | 32.06/23.06 | 31.22/24.29 | 44.61/41.39 | 46.68/44.83 | 48.86/39.14 | **61.33/50.29** |
| | | Acc | 23.62/18.91 | 33.73/26.12 | 38.18/27.94 | 39.94/26.24 | 49.58/41.61 | | | **65.46/60.37** |
| Gowalla | TUL | $F_1$ | 6.36/1.21 | 18.87/12.20 | 19.51/16.15 | 25.84/14.72 | 29.71/20.50 | 36.06/24.99 | 36.74/33.68 | **44.50/38.76** |
| | | Acc | 16.18/5.38 | 23.28/16.45 | 24.16/26.09 | 29.28/28.15 | 31.42/20.75 | 37.92/26.94 | 37.58/35.48 | **49.60/44.48** |
| SHL | TMI | $F_1$ | 44.23/40.24 | 40.91/38.66 | 47.47/36.50 | 46.33/36.50 | 52.88/49.23 | 56.88/51.31 | 59.83/51.59 | **71.54/63.67** |
| | | Acc | 55.92/51.15 | 49.24/49.69 | 54.06/45.64 | 53.21/45.14 | 59.70/54.82 | 62.16/55.88 | 65.72/56.97 | **74.34/65.30** |
| | TSeg | $F_1$ | – | 15.30/14.39 | 24.80/18.43 | 29.40/20.65 | 36.42/32.09 | 42.26/32.82 | 42.76/32.45 | **47.15/46.17** |
| | | Acc | | 23.77/18.14 | 29.14/21.16 | 30.36/28.69 | 39.98/33.00 | 43.21/37.51 | 45.42/36.95 | **53.75/51.17** |

[1] The percent sign of the $F_1$ score and accuracy results is omitted.
[2] S-TDP refers to various SOTA methods in a single trajectory data preparation task.

This demonstrates that simply training separate models for individual tasks cannot capture the shared spatio-temporal knowledge needed for generalized trajectory reasoning. Second, although LLM-based tabular data preparation methods exhibit moderate zero-shot performance due to the base LLM's inherent generalization ability, they still fall short. In contrast, FedTDP improves performance by at least **32.26%**. The gap stems from the fact that generic tabular LLMs treat trajectory data as unordered rows, overlooking the sequential, temporal, and spatial dependencies essential for mobility modeling. Third, compared with SOTA LLM-based spatio-temporal analysis methods, FedTDP achieves improvements ranging from **4.84%** to **45.22%**. These methods do not incorporate TDP-specific knowledge and cannot adapt to the diverse semantics required by 10 heterogeneous TDP tasks. In contrast, FedTDP leverages the Trajectory Knowledge Enhancer (TKE) to inject trajectory-aware priors and unify task instructions, enabling both the SLM and LLM to perform robust reasoning across unseen tasks and unseen domains. Overall, the results confirm that the base LLM alone provides limited benefits, and the substantial performance gains arise from FedTDP's trajectory-aware knowledge enhancement and federated trajectory learning design.

## 5.2 ABLATION STUDY

We evaluate the effectiveness of each module by systematically removing one at a time, yielding FedTDP without TPA (w/o TPA), without TKE (w/o TKE), and without FPO (w/o FPO). The results are shown in Fig. 5. First, the performance of FedTDP is slightly degraded compared to w/o TPA, as TPA can not fully capture the spatio-temporal information of the trajectory data, leading to a marginal performance decline when using TPA. Besides, FedTDP has a little in runtime and communication costs with approximately 5GB additional communication over 10 rounds, because TPA transmits higher-dimensional embedding data instead of three-dimensional spatio-temporal points. However, to safeguard data privacy, the use of TPA in the FedTDP framework is essential. Second, removing TKE causes a dramatic performance drop with at least **27.52%** degradation, highlighting that the base LLM's inherent generalization ability is insufficient for TDP reasoning. TKE introduces trajectory-aware prompting, offsite-tuning, sparse-tuning, and bidirectional knowledge distillation, all of which guide the LLM/SLM to acquire TDP-specific semantics. Notably, models without TKE also incur higher training costs because they must update more parameters. These results confirm that TKE is the primary source of FedTDP's strong cross-task and cross-dataset generalization. Finally, the performance does not change significantly compared to w/o FPT, but its training runtime and communication overhead are significantly reduced by almost **4 times** less. This reduction is because FPO can reduce data transmission and improve training efficiency.

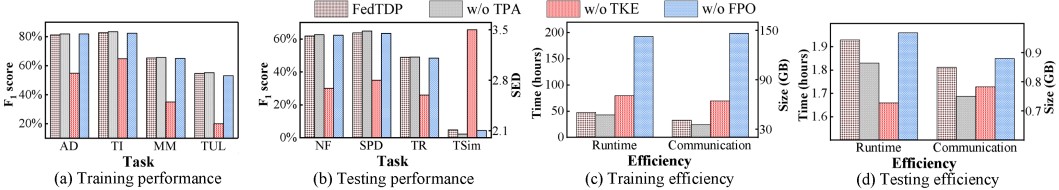

Figure 5: The ablation study

## 5.3 MODEL GENERALIZATION STUDY

To evaluate the proposed FedTDP framework generalization in different numbers of training tasks, we systematically remove the training task from back to front based on their order in Table 1. As illustrated in Fig. 6, the results indicate that the performance of FedTDP across various tasks declines as the number of training tasks decreases. This decline is primarily

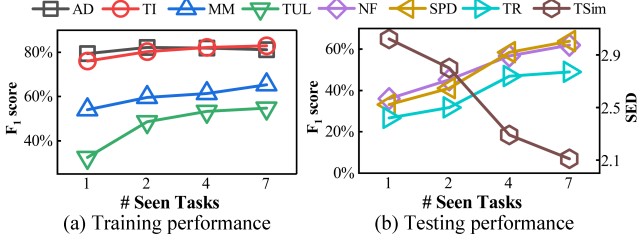

Figure 6: Generalization study

attributed to the reduced acquisition of TDP knowledge, which adversely affects generalization. Notably, when the number of tasks is reduced to one (i.e., training FedTDP solely on the anomaly detection task using GeoLife), the performance also falls below that of S-TDP in Table 2.

## 5.4 EFFICIENCY STUDY

Fig. 7 shows the communication costs (in GB) and running times (in hours) of various methods across all TDP tasks. During the training process, the proposed framework incurs the largest communication size because it must transfer embeddings and model parameters, whereas LLM-based methods transmit all perturbed data, generated through differential privacy, to

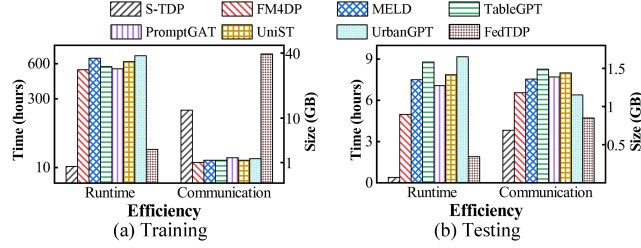

Figure 7: Efficiency study

the server in the first round of training and do not require data transmission in the following rounds of training. Furthermore, the runtime of FedTDP is reduced by a factor of **11.3 to 14.2** compared to other LLM-based methods, effectively mitigating the high computational costs typically incurred by LLMs due to their massive parameter sizes. In the testing phase, the communication size of FedTDP is nearly identical to that of S-TDP and **1.4 to 1.8** times less than that of other LLM-based methods, which require transferring all data to the server, whereas FedTDP only transmits the data necessary for cross-client TDP. Additionally, the runtime of FedTDP is **2.6 to 4.8** times lower than that of other LLM-based methods, further underscoring its efficiency.

## 5.5 ADDITIONAL EXPERIMENTS

We further evaluate FedTDP across the following aspects. We evaluate the performance of different LLMs and SLMs to examine the influence of backbone choice in **Appendix D.4**. We aim to study the robustness under hyperparameter settings in **Appendix D.5**.

## 6 CONCLUSION

This paper presented FedTDP, a unified federated framework for trajectory data preparation (TDP). FedTDP integrates three key components: (1) a trajectory privacy autoencoder designed to protect sensitive data while preserving spatio-temporal correlations; (2) a trajectory knowledge enhancer that develops TDP-oriented large language models (LLMs); and (3) a parallel optimization mechanism to boost computational efficiency. Extensive experiments on six datasets across ten TDP tasks demonstrated the framework's superior effectiveness, efficiency, and robustness compared to state-of-the-art methods. Limitations and future directions of this paper are discussed in **Appendix E**.

## 7 ACKNOWLEDGMENT

This work was supported in part by the NSFC under Grants No. (62402422, 62025206, U23A20296, and 62472377), Yongjiang Talent Introduction Programme (2024A-162-G), Zhejiang Provincial Natural Science Foundation of China under Grant No. LZ25F020001, and Zhejiang Province's "Lingyan" R&D Project under Grant No. 2024C01259. Ziquan Fang is the corresponding author.

## ETHICS STATEMENT

We affirm that this work fully adheres to the ICLR Code of Ethics. We do not collect, use, or release any new human subject data. All experiments are conducted on established and publicly available datasets, which have been widely used in prior academic research.

## REPRODUCIBILITY STATEMENT

We are committed to ensuring the full reproducibility of our work. All source code, datasets, and detailed experimental configurations are publicly available at `https://github.com/ZJU-DAILY/FedTDP`. Section 5 provides a comprehensive overview of the evaluation setup, including the specific datasets and baseline methods. For theoretical claims, complete proofs for Theorems 1 and 2, along with the theoretical privacy analysis, are provided in Appendices C.1, C.4, and C.2. Detailed descriptions of all datasets, including their statistics and the specific quality issues they exhibit, are found in Appendix D.1. Detailed descriptions of all baselines are found in Appendix D.2. Details of evaluation metrics and hardware environment are provided in Appendix D.3.

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

APPENDIX

APPENDIX

In the subsequent sections, we present supplementary materials to provide more details of this paper, offering deeper insights and additional technical details for readers seeking further clarification. The appendix is organized as follows.

In **Section A**, we present a systematic review of related work to help readers understand the key development in areas relevant to this paper, including (i) the latest trajectory data preparation methods, (ii) applications of large language models in other data preparation, and (iii) recent advancements in large language models for spatio-temporal data analysis.

In **Section B**, we summarize the preliminary of notations and trajectory data preparation tasks for better understanding our work, including (i) the frequently used notations and (ii) detailed description of trajectory data preparation tasks.

In **Section C**, we provide the additional methodology details to support the analysis shown in the main body of this paper, including (i) complete theoretical proofs of Theorems 1 and 2, (ii) practical examples of trajectory data preparation tasks using the trajectory prompt engineering of the proposed trajectory knowledge enhancer, and (iii) the training process of FedTDP.

In **Section D**, we describe the extensive experimental details to provide more information about experimental settings and further demonstrate the superior performance of the proposed FedTDP framework, including (i) datasets description, (ii) compared baselines introduction, (iii) evaluation metrics and implementations details, and (iv) the detailed experimental results of model base and parameter sensitivity studies.

In **Section E**, we discuss several limitations of the proposed FedTDP framework that warrant further exploration.

In **Section F**, we provide a transparent account of the role Large Language Models (LLMs) played in the preparation of this manuscript, in accordance with ethical research practices and ICLR's commitment to research integrity.

## A  RELATED WORK

### A.1  TRAJECTORY DATA PREPARATION

Numerous Trajectory Data Preparation (TDP) methods have been proposed to improve the quality of trajectory data for trajectory data preparation tasks. For the **anomaly detection** task, ATROM (Gao et al., 2023) addresses the critical challenge of anomaly recognition in open-world scenarios through the development of a probabilistic metric learning model, which significantly improves the accuracy of anomaly detection in complex environments. For the **trajectory imputation** task, Kamel (Musleh & Mokbel, 2023) proposes a scalable architecture that incorporates additional real trajectory points to predict the missing trajectory data, improving the accuracy of trajectory imputation. For the **map matching** task, GraphMM (Liu et al., 2024b) leverages the graphical structure using a graph neural network to effectively model the topology of road network and trajectory features, improving the accuracy of map matching. For the **trajectory-user link** task, AttnTUL (Chen et al., 2024b) introduces a hierarchical spatio-temporal attention neural network, which co-encodes local trajectory transition patterns and global spatial dependencies to establish links between trajectories and users more accurately. For the **travel mode identification** task, Estimator (Hu et al., 2024a) proposes an effective and scalable framework that partitions the traffic space into disjoint spatial regions based on traffic conditions, improving the accuracy of travel mode identification. For the **trajectory simplification** task, S3 (Fang et al., 2023) presents a lightweight framework consisting of two chained sequence-to-sequence modules, which is integrated within a graph neural architecture, improving the accuracy and efficiency of trajectory simplification. For the **trajectory recovery** task, LightTR (Liu et al., 2024c) presents an efficient framework using a local trajectory embedding module, robust feature extraction capabilities while significantly reducing computational overhead.

However, none of these studies have considered data privacy constraints. They typically assume that the trajectory data collection is centralized, which introduces a significant risk of privacy leakage, especially in federated learning environments. In addition, all of them are single-task methods. When

Table 3: Notation and description

| Notation | Description |
|---|---|
| $p$ | A spatio-temporal point consisting of location and time $\langle l, t \rangle$ |
| $T$ | A trajectory consisting of multiple spatio-temporal points $\{p_1, p_2, \ldots\}$ |
| $ST$ | A sub-trajectory of the trajectory $T$ |
| $S$ | A data silo that represents a region |
| $C$ | A client that represents a region |
| $D$ | The trajectory database in a client $C$ |
| $\mathcal{C}$ | A set of clients $\{C_1, C_2, \ldots\}$ |
| $\mathcal{D}$ | A set of trajectory datasets $\{D_1, D_2, \ldots\}$ |
| $\theta_{LLM}, \theta_{SLM}$ | The large language model and small language model |

handling multiple TDP tasks, different models need to be trained for each specific task. It not only demands substantial time and computational resources but also results in poor model generalization ability. ***In contrast, we aim to propose a unified federated framework to support various trajectory data preparation tasks while safeguarding trajectory data privacy.***

## A.2 LARGE LANGUAGE MODELS FOR OTHER DATA PREPARATION

A few works on table data preparation using Large Language Models (LLMs) have been proposed recently. For instance, MELD (Yan et al., 2024) introduces a general solver for low-resource table data preparation, leveraging a mixture-of-experts architecture to support merging and augmentation of domain-specific experts trained on limited annotated examples. Besides, TableGPT (Li et al., 2024a) presents a table-tuning paradigm, where LLMs are fine-tuned using various table tasks synthesized from real tables to enhance the model's ability to understand and process table-related tasks. Moreover, Narayan et al. (2022) explores the performance of LLMs for table data preparation tasks, which evaluates their performance on five data cleaning and integration tasks through prompt-based methods.

However, these works are specifically tailored for table data preparation and are not directly applicable to trajectory data preparation tasks. They lack the necessary understanding of trajectory data and do not account for the spatio-temporal characteristics and complexity of trajectory data preparation tasks, making them unsuitable for such applications. ***In contrast, we aim to develop a TDP-oriented LLM to effectively support various trajectory data preparation tasks.***

## A.3 LARGE LANGUAGE MODELS FOR SPATIO-TEMPORAL DATA ANALYSIS

There are a few spatio-temporal LLMs proposed (Li et al., 2024c; Zhang et al., 2023b; Li et al., 2023), which have achieved superior performance in spatio-temporal downstream applications. For example, UrbanGPT (Li et al., 2024b) integrates a spatio-temporal dependency encoder with a command adjustment paradigm to enhance the LLMs' understanding of complex temporal and spatial interdependencies. Besides, UniST (Yuan et al., 2024) develops a general-purpose model for urban spatio-temporal prediction through diverse data utilization, effective pre-training, and knowledge-guided prompts. In addition, PromptGAT (Da et al., 2024) employs prompt-based grounded action transformations to analyze system dynamics by leveraging reasoning capabilities of large language models to understand environmental impacts on traffic patterns.

However, none of them have considered trajectory data quality. If the quality of trajectory data is extremely poor, the performance of spatio-temporal large language models in downstream tasks will not be satisfactory either. ***In contrast, we aim to explore the powerful capabilities of large language models for trajectory data preparation to enhance the quality of trajectory data.***

## B NOTATION AND TRAJECTORY DATA PREPARATION TASK

**Notation and Description.** We first present the frequently used notations and descriptions in this paper, as listed in Table 3.

Table 4: Trajectory data preparation task

| Task Category | Task | Description |
|---|---|---|
| Data Cleaning | Anomaly Detection (AD) | Detect anomalous trajectory |
| | Trajectory Imputation (TI) | Predict missing points |
| | Noise Filtering (NF) | Filter point noise |
| | Stay Point Detection (SPD) | Identify stationary points |
| Data Matching | Map Matching (MM) | Align a trajectory to road network |
| | Trajectory-User Linking (TUL) | Associate trajectories with users |
| Data Annotation | Travel Mode Identification (TMI) | Identify transportation mode |
| Data Reduction | Trajectory Simplification (TSim) | Remove number of points |
| | Trajectory Segmentation (TSeg) | Divide a trajectory to segments |
| Data Augmentation | Trajectory Recovery (TR) | Recovery complete trajectory |

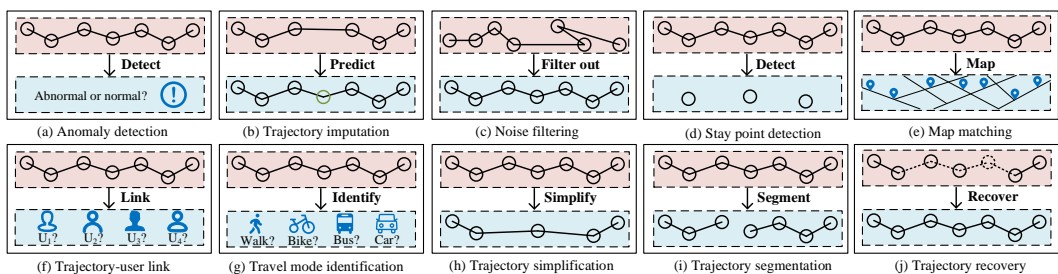

Figure 8: The supported trajectory data preparation tasks

**Trajectory Data Preparation Task.** Besides, we summarize the supported trajectory data preparation tasks of the proposed FedTDP in this paper, as shown in Table 4, with the rough processing of each task shown in Fig. 8.

# C ADDITIONAL METHODOLOGY DETAILS

## C.1 PROOF OF THEOREM 1

We provide the complete theoretical proof of Theorem 1 proposed in this paper that provides the correctness of the trajectory privacy autoencoder model aggregation, as detailed below.

*Proof.* According to Eq. 1, we can get the result of aggregating masked parameter blocks $\{\tilde{P}_1^{(k)}, \tilde{P}_2^{(k)}, \ldots, \tilde{P}_{|\mathcal{C}|}^{(k)}\}$ for all clients, as formally shown below:

$$\sum_{i=1}^{|\mathcal{C}|} \tilde{P}_i^{(k)} = \sum_{i=1}^{|\mathcal{C}|} P_i^{(k)} + \sum_{i=1}^{|\mathcal{C}|} \sum_{j=1 \& j \neq i}^{|\mathcal{C}|} a_{i,j} * sk_{i,j} \tag{10}$$

Here, $sk_{i,j} = sk_{j,i}$ and $a_{i,j} = -a_{j,i}$, thus we can get the result formally shown below:

$$\sum_{i=1}^{|\mathcal{C}|} \sum_{j=1 \& j \neq i}^{|\mathcal{C}|} a_{i,j} * sk_{i,j} = \sum_{i=j+1}^{|\mathcal{C}|} \sum_{j=1}^{|\mathcal{C}|} (a_{i,j} * sk_{i,j} + a_{j,i} * sk_{j,i}) = 0 \tag{11}$$

The aggregation $\tilde{P}^{(k)}$ of the masked parameter block is formally shown below:

$$\sum_{i=1}^{|\mathcal{C}|} \tilde{P}_i^{(k)} = \sum_{i=1}^{|\mathcal{C}|} P_i^{(k)} + \sum_{i=1}^{|\mathcal{C}|} \sum_{j=1 \& j \neq i}^{|\mathcal{C}|} a_{i,j} * sk_{i,j} = \sum_{i=1}^{|\mathcal{C}|} P_i^{(k)} \tag{12}$$

$\square$

## C.2 THEORETICAL PRIVACY ANALYSIS

The privacy protection mechanism of the proposed FedTDP framework is built upon the Trajectory Privacy Autoencoder (TPA), which protects trajectory data privacy while maintaining spatio-temporal correlations. Besides, it develops a decentralized aggregation approach based on secret sharing (Shamir, 1979) that ensures the parameters of the TPA model remain confidential against trajectory data recovery or inference through embedding (Song & Raghunathan, 2020; Chen et al., 2024c; Huang et al., 2024) and gradient (Wang et al., 2024; Zheng et al., 2024; Zhu et al., 2023) inversion attacks. Specifically, the privacy of our framework is rigorously protected through two safeguards to defend against advanced embedding and gradient inversion attacks:

- To prevent embedding inversion attacks, we ensure that the TPA model is kept private and never exposed to the server. The server only receives encoded embeddings without access to the embedding model parameters, thereby breaking the necessary precondition for embedding inversion. Without knowledge of the embedding model, reconstructing raw trajectories from embeddings is infeasible.
- To prevent gradient inversion attacks during TPA model aggregation, we design a decentralized secret-sharing protocol for secure model aggregation. Instead of exchanging raw gradients, the TPA model is split into several parameter blocks that are then masked using pairwise secret keys before transmission. Then, each client is responsible for aggregating one block without access to other clients' model updates, ensuring that no client can reconstruct another client's data from the gradients. The correctness of this aggregation scheme is formally proven in Appendix C.1.

Overall, our privacy guarantee stems from isolating the TPA model from the server and protecting the TPA model during the federated optimization process. To rigorously analyze the privacy-preserving capability of TPA, we first define the threat model as follows.

**Threat Model**. Following prior works (Zhang et al., 2024c; Tong et al., 2025; Zhao et al., 2024) in federated learning, we assume the server acts as a semi-honest adversary who will honestly execute required operations (e.g., aggregation) but also remains curious about the private client data. In the F-TDP problem, the server seeks to reconstruct clients' raw trajectory data using adversary's knowledge, which includes the client model architecture, including the client model architecture, the privacy-preserving mechanism, and the encoded embeddings uploaded by clients.

Based on this, we use mutual information (Kreer, 1957) to quantify the upper bound of privacy leakage, i.e., $I(T; E)$, which measures the information about original trajectory data $T$ that can be inferred from encoded embeddings $E$ transmitted to the server, as shown below:

$$I(T; E) = H(T) - H(T|E), \tag{13}$$

where $H(\cdot)$ denotes the entropy. Since $E$ is derived from $T$ through the encoder $\theta_{Enc}$, the conditional entropy $H(T|E)$ can be decomposed as:

$$H(T|E) = \mathbb{E}_{\theta_{Enc} \sim \mathcal{P}_\Theta}[H(T|E, \theta_{Enc})] + H(\theta_{Enc}|E), \tag{14}$$

where $\theta_{Enc}$ is drawn from a prior distribution $\mathcal{P}_\Theta$ and $\Theta$ is the parameter space. Besides, both $H(T|E, \theta_{Enc})$ and $H(\theta_{Enc}|E)$ are large because $\theta_{Enc}$ is private and inaccessible to the server. Consequently, the conditional entropy $H(T|E)$ remains high, leading to minimal privacy leakage $I(T; E)$. Furthermore, leveraging Bayes' theorem (Bayes, 1958) and Fano's inequality (Fano & Hawkins, 1961), the probability $P_e$ of the attacker recovering $T$ incorrectly satisfies:

$$H(P_e) + P_e log|\mathcal{T}| \geq H(T|E), \tag{15}$$

where $\mathcal{T}$ denotes the trajectory data space. The large $H(T|E)$ results in a correspondingly high $P_e$, indicating a low likelihood of successful reconstruction of the original trajectory data, which further underscores the effectiveness of TPA in protecting trajectory privacy.

## C.3 EXAMPLES OF TRAJECTORY PROMPT ENGINEERING

For better understanding the trajectory prompt engineering of the proposed trajectory knowledge enhancer, we show practical examples of noise filtering and travel mode identification tasks using the trajectory prompt engineering in the small language model of the client.

As shown in Fig. 9, *Task* shows the task name with its description listed in Section 3. Besides, *Data* uses the raw trajectory data in clients, consisting of spatio-temporal points i.e., $T = \{p_1, p_2, \ldots\}$.

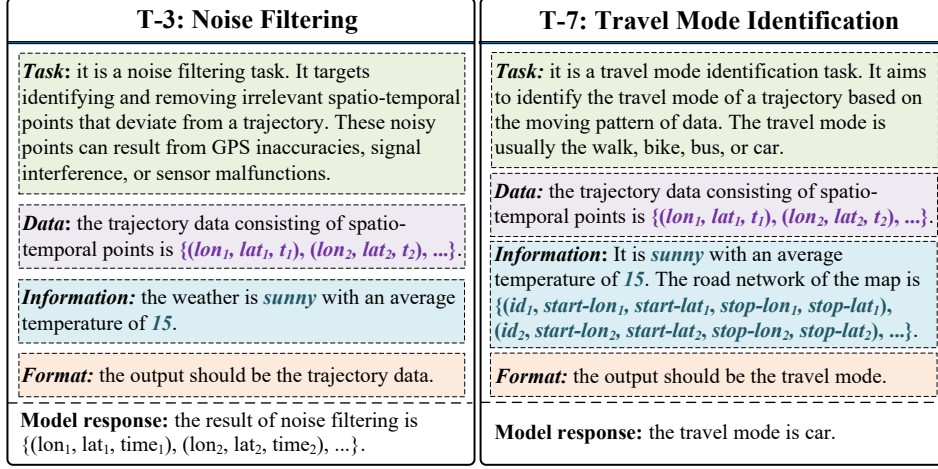

| T-3: Noise Filtering | T-7: Travel Mode Identification |
|---|---|
| **Task**: it is a noise filtering task. It targets identifying and removing irrelevant spatio-temporal points that deviate from a trajectory. These noisy points can result from GPS inaccuracies, signal interference, or sensor malfunctions. | **Task:** it is a travel mode identification task. It aims to identify the travel mode of a trajectory based on the moving pattern of data. The travel mode is usually the walk, bike, bus, or car. |
| **Data**: the trajectory data consisting of spatio-temporal points is {(*lon₁, lat₁, t₁*), (*lon₂, lat₂, t₂*), ...}. | **Data:** the trajectory data consisting of spatio-temporal points is {(*lon₁, lat₁, t₁*), (*lon₂, lat₂, t₂*), ...}. |
| **Information:** the weather is *sunny* with an average temperature of *15*. | **Information**: It is *sunny* with an average temperature of *15*. The road network of the map is {(*id₁, start-lon₁, start-lat₁, stop-lon₁, stop-lat₁*), (*id₂, start-lon₂, start-lat₂, stop-lon₂, stop-lat₂*), ...}. |
| **Format:** the output should be the trajectory data. | **Format:** the output should be the travel mode. |
| **Model response:** the result of noise filtering is {(lon₁, lat₁, time₁), (lon₂, lat₂, time₂), ...}. | **Model response:** the travel mode is car. |

Figure 9: The example of trajectory prompt engineering in clients

Additionally, ***Information*** includes optional road network and weather data. Specifically, *sunny* and 15 represent the weather tokens during the trajectory, *id* denotes the segment ID, *start-lon* and *start-lat* indicate the segment's longitude and latitude at the starting point, and *stop-lon* and *stop-lat* denote the segment's longitude and latitude at the stopping point. Specifically, we utilize the weather data for these tasks and the road network for the travel mode identification task. Moreover, ***Format*** refers to the expected output data of the task. We anticipate the outputs of noise filtering and travel mode identification to be the trajectory data and travel mode, respectively. Finally, the model's response should return the expected results in ***Format***. Note that, in the server ***Data*** uses embeddings uploaded by clients, consisting of encoded spatio-temporal points i.e., $E = \{e_1, e_2, \ldots\}$.

Note that the inclusion of the "Information" field in the Trajectory Knowledge Enhancer (TKE) is task-specific and manually configured based on domain knowledge, rather than automatically determined by the model. For instance, road network information is indispensable for the Map Matching task, as understanding road topology is crucial. In contrast, weather information is optional and only incorporated when external conditions are expected to affect mobility behavior. For example, in the Travel Mode Identification task, weather can influence the choice of transportation mode. This design ensures flexibility, enabling the integration of the most relevant contextual factors for each TDP task while avoiding unnecessary noise.

## C.4 PROOF OF THEOREM 2

We provide the complete proof for Theorem 2, which gives the probability that a layer $L_i$ is selected during LoRA sparse-tuning when $M$ layers are sampled without replacement according to their normalized change ratios $R(L_j)$.

*Proof.* We consider the standard weighted sampling without replacement procedure. Let $R(L_j)$ denote the probability of selecting layer $L_j$ at the first draw. After each draw, the selected layer is removed and the remaining probabilities are renormalized.

**Step 1: Probability of selecting $L_i$ at the first draw.** Since the first draw is directly sampled according to $R(L_j)$, we have

$$P_1(L_i) = R(L_i) \tag{16}$$

**Step 2: Probability of selecting $L_i$ at the second draw.** For $L_i$ to be selected at the second draw, some layer $L_{j_1}$ ($j_1 \neq i$) must be drawn first, after which $L_i$ is sampled from the remaining layers. The probability of this event is

$$R(L_{j_1}) \cdot \frac{R(L_i)}{1 - R(L_{j_1})} \tag{17}$$

Summing over all possible $j_1 \neq i$, we obtain

$$P_2(L_i) = R(L_i) \sum_{\substack{j_1 \\ j_1 \neq i}}^{N} \frac{R(L_{j_1})}{1 - R(L_{j_1})} \tag{18}$$

**Step 3: Probability of selecting $L_i$ at the third draw.** For $L_i$ to be selected at the third draw, two distinct layers $L_{j_1}$ and $L_{j_2}$ ( $j_1 \neq j_2 \neq i$) must be drawn first, in any order. The probability of a specific sequence $(j_1, j_2)$ followed by $i$ is

$$R(L_{j_1}) \cdot \frac{R(L_{j_2})}{1 - R(L_{j_1})} \cdot \frac{R(L_i)}{1 - R(L_{j_1}) - R(L_{j_2})}. \tag{19}$$

Thus,

$$P_3(L_i) = R(L_i) \sum_{\substack{j_1, j_2 \\ j_1 \neq j_2 \neq i}}^{N} \frac{R(L_{j_1})R(L_{j_2})}{(1 - R(L_{j_1}))(1 - R(L_{j_1}) - R(L_{j_2}))} \tag{20}$$

**Recursive step: Probability of selecting $L_i$ at the $M$-th draw.** Extending the above pattern, for $L_i$ to be selected exactly at draw $M$, the first $M-1$ draws must be any sequence of distinct layers

$$(j_1, \ldots, j_{M-1}), \qquad j_t \neq i, \ j_a \neq j_b, \tag{21}$$

and the $M$-th draw selects $L_i$ from the remaining probability mass. Thus, the probability of one such sequence is

$$R(L_{j_1}) \prod_{t=2}^{M-1} \frac{R(L_{j_t})}{1 - \sum_{s=1}^{t-1} R(L_{j_s})} \cdot \frac{R(L_i)}{1 - \sum_{s=1}^{M-1} R(L_{j_s})} \tag{22}$$

Therefore, the probability of selecting $L_i$ at draw $M$ is

$$P_M(L_i) = R(L_i) \sum_{\substack{j_1, \ldots, j_{M-1} \\ j_t \neq i, \ j_a \neq j_b}}^{N} \frac{R(L_{j_1}) \cdots R(L_{j_{M-1}})}{(1 - R(L_{j_1})) \cdots \left(1 - \sum_{t=1}^{M-1} R(L_{j_t})\right)} \tag{23}$$

**Final Expression.** The probability that $L_i$ is selected in any of the $M$ draws is thus

$$P(L_i, M) = \sum_{j=1}^{M} P_j(L_i) = R(L_i) + R(L_i) \sum_{\substack{j_1 \\ j_1 \neq i}}^{N} \frac{R(L_{j_1})}{1 - R(L_{j_1})} + \cdots$$

$$+ R(L_i) \sum_{\substack{j_1, \ldots, j_{M-1} \\ j_t \neq i, \ j_a \neq j_b}}^{N} \frac{R(L_{j_1}) \cdots R(L_{j_{M-1}})}{(1 - R(L_{j_1})) \cdots \left(1 - \sum_{t=1}^{M-1} R(L_{j_t})\right)} \tag{24}$$

**Correctness Verification.** We finally verify that the derived probabilities satisfy $\sum_{i=1}^{N} P(L_i, M) = M$, i.e., the expected number of selected layers is exactly $M$.

Recall that $P_k(L_i)$ denotes the probability that layer $L_i$ is selected at the $k$-th draw. By definition,

$$P(L_i, M) = \sum_{k=1}^{M} P_k(L_i),$$

and thus

$$\sum_{i=1}^{N} P(L_i, M) = \sum_{i=1}^{N} \sum_{k=1}^{M} P_k(L_i) = \sum_{k=1}^{M} \sum_{i=1}^{N} P_k(L_i).$$

It therefore suffices to show that for each draw $k$, $\sum_{i=1}^{N} P_k(L_i) = 1$. Fix a particular draw index $k$ and consider any concrete sequence of previously selected layers $(j_1, \ldots, j_{k-1})$. Conditional on this sequence, the probability of selecting layer $L_i$ at step $k$ is

$$P_k(L_i) = \frac{R(L_i)}{1 - \sum_{t=1}^{k-1} R(L_{j_t})},$$

---

**Algorithm 1:** The training on the server

---

**Input:** the number of training rounds *TR*

1 **for** *round $r = 0, \ldots, TR - 1$* **do**
2     $f \leftarrow$ *IsFrozen*$(r)$ // Get the frozen status of this round;
3     **if** $f ==$ `False` **then**
4        $E \leftarrow (\mathcal{C})$ // Get the embeddings data from clients;
5        $E \leftarrow$ *Connect*$(E)$ // Connect into a complete embeddings;
6     **else**
7        $E \leftarrow$ *GetFrozenData*$(r - 1)$ // Get the frozen data;
8     *prompt* $\leftarrow$ *TKE*$(E)$ // Construct the prompt of the embeddings;
9     $o \leftarrow \theta_{LLM}($*prompt*$)$ // Input the prompt data to the LLM;
10    $o \leftarrow$ *Split*$(o)$ // Split the output of the LLM;
11    **if** $f ==$ `False` **then**
12       **for** *client number $i = 0, \ldots, |\mathcal{C}| - 1$* **do**
13         *Send*$(o_i, C_i)$ // Send split results to respective clients;

---

for any $i$ that has not been selected in the first $k - 1$ draws. Summing over all remaining layers, we obtain

$$\sum_{i \notin \{j_1, \ldots, j_{k-1}\}} \frac{R(L_i)}{1 - \sum_{t=1}^{k-1} R(L_{j_t})} = \frac{\sum_{i \notin \{j_1, \ldots, j_{k-1}\}} R(L_i)}{1 - \sum_{t=1}^{k-1} R(L_{j_t})} = \frac{1 - \sum_{t=1}^{k-1} R(L_{j_t})}{1 - \sum_{t=1}^{k-1} R(L_{j_t})} = 1.$$

In other words, for any fixed history $(j_1, \ldots, j_{k-1})$, the denominator $1 - \sum_{t=1}^{k-1} R(L_{j_t})$ is exactly canceled by the sum of the remaining $R(L_i)$. Averaging over all possible histories of the first $k - 1$ draws does not change this fact, and we still have $\sum_{i=1}^{N} P_k(L_i) = 1$. Therefore,

$$\sum_{i=1}^{N} P(L_i, M) = \sum_{k=1}^{M} \sum_{i=1}^{N} P_k(L_i) = \sum_{k=1}^{M} 1 = M,$$

which shows that the expected number of selected layers is indeed $M$, as desired. $\qquad\square$

### C.5   Multi-Task Training

Due to the diverse range of trajectory data preparation tasks that need to be addressed, we propose a multi-task training strategy to enhance the model's learning and generalization capabilities. Specifically, we prepare a trajectory dataset applicable to most trajectory data preparation tasks and construct labels for each task. During the training phase, we execute multiple trajectory data preparation tasks on the same trajectory data input, calculate the loss for each task, and jointly optimize the model formulaically shown below:

$$\mathcal{L} = \mathcal{L}_{\text{T-1}} + \mathcal{L}_{\text{T-2}} + \ldots + \mathcal{L}_{\text{T-10}}, \tag{25}$$

where $\mathcal{L}_{T\text{-}i}$ is the loss of trajectory data preparation task T-$i$ listed in the Section 3. Note that the proposed FedTDP framework can be easily extended to support other trajectory data preparation tasks benefiting from its modular architecture, decoupled data processing pipeline, and variable model base.

**Training Algorithm.** For convenient method reproduction, we provide a detailed training process of the entire FedTDP framework, which can be divided into the server and client, as shown in Algorithms 1 and 2.

In the server (i.e., Algorithm 1), the input is the number of training rounds (line 1). For each training round $r$, it begins to get the frozen state $f$ (lines 2–3). If $f$ is not frozen, the server gets the trajectory embeddings $E$ from clients $\mathcal{C}$ and connects them, or it gets local $E$ frozen in the last training round $r - 1$ (lines 4–9). Then, the server uses TKE to construct the TDP prompt for the LLM and get the output $o$ (lines 10–11). Finally, the server splits it into several parts and sends them to respective clients if $f$ is not frozen (lines 12–18).

---

**Algorithm 2:** The training on the client

---

**Input:** the number of training rounds *TR* and server *s*

1  **for** *round* $r = 0, \ldots, TR - 1$ **do**
2      $D \leftarrow GetData()$ // Get the local trajectory data;
3      $f \leftarrow IsFrozen(r)$ // Get the frozen status of this round;
4      **if** $f == False$ **then**
5          $E \leftarrow Enc(D)$ // Encode the trajectory into embeddings;
6          $Send(E, s)$ // Send the embeddings data to the server;
7      $prompt \leftarrow TKE(D)$ // Construct the prompt of the data;
8      $o' \leftarrow \theta_{SLM}(prompt)$ // Input the prompt data to the SLM;
9      **if** $f == False$ **then**
10         $o \leftarrow Get(s)$ // Get the result from the server;
11         $o \leftarrow Dec(o)$ // Decode the server's result;
12     **else**
13         $o \leftarrow GetFrozenData(r - 1)$ // Get the frozen data;
14     $result \leftarrow TKE(o', o)$ // Compute the distillation result;

---

In the client (i.e., Algorithm 2), the input are the number of training rounds and the server (line 1). For each training round $r$, it begins to get the trajectory data $D$ and frozen state $f$ (lines 2–4). If $f$ is not frozen, clients encode $D$ into embeddings and send them to the server (lines 5–8). Then, clients use TKE to construct the TDP prompt for the SLM and get the output $o'$ (lines 9–10). If $f$ is not frozen, clients get the LLM's output $o$ from the server and decode it, or it gets local $o$ frozen in the last round $r - 1$ (lines 11–16). Finally, clients use TKE to compute the final result between $o'$ and $o$ (lines 17–18).

**Complexity Analyses.** We also give complexity analyses for Algorithms 1 and 2. Specifically, given the number of trajectory embeddings data $|E|$ from all clients, the complexity of Algorithm 1 is $O(|E| * TR * MC)$, where *MC* is the model complexity of the LLM. Given the number of trajectories $|D|$ in the client, the complexity of Algorithm 2 is $O(|D| * TR * MC')$, where $MC'$ is the model complexity of the SLM.

### C.6 EXAMPLES OF LLM/SLM ON TRAJECTORY DATA PREPARATION TASK

To better explain what the LLM/SLM actually learns, we provide two qualitative examples of LLM/SLM outputs on local trajectory simplification task and cross-client trajectory imputation tasks.

**Local Trajectory Simplification Task.** For a local trajectory simplification task, the client's SLM processes a sub-trajectory, as shown in Fig. 10. Specifically, (i) the raw data is first normalized using MinMax scaling, (ii) the client uses the Trajectory Knowledge Enhancer (TKE) to construct a structured prompt, and (iii) the client's SLM processes this prompt and output the simplification result, where shows that it removed the second and last points.

**Cross-Client Trajectory Imputation Task.** For a cross-client trajectory imputation task, two clients hold sequential sub-trajectories of the same user, as shown in Fig. 11. Specifically, (i) these two clients first normalized their local raw data, respectively. (ii) each client uses the TPA to encode its sub-trajectory into embeddings sent to the server (For brevity, we denote these embeddings as $[p1, p2, p3, ()]$ for the first client and $[p5, p6, p7]$ for the second client.), (iii) the server aligns the embeddings and use TKE to construct a prompt, (iv) the server's LLM processes this prompt and output the imputation result, (v) the result is split and sent back to clients, and (vi) the first client uses TPA to decode the server's output for p4: $(0.2021, 0.707531, 0.612032)$, with the original format: $(1201957233, 116.56446, 39.91442)$.

---

**Local Trajectory Simplification**

---

*Processing*

**Raw Data:** [(1201963833, 116.69167, 39.85174), (1201964432, 116.69167, 39.85175), (1201965032, 116.69167, 39.85176), (1201965632, 116.69172, 39.85208), (1201965632, 116.69172, 39.85208), (1201966232, 116.69172, 39.85199), (1201966832, 116.69171, 39.85196), (1201967432, 116.69171, 39.85182)]

**Normalized Data:** [(0.2516, 0.772114, 0.576668), (0.2561, 0.772114, 0.576674), (0.2606, 0.772114, 0.576679), (0.2651, 0.772140, 0.576860), (0.2651, 0.772140, 0.576860), (0.2696, 0.772140, 0.576809), (0.2741, 0.772135, 0.576792), (0.2786, 0.772135, 0.576713)]

---

*Prompt*

**Task:** it is a trajectory simplification task. It aims to reduce the number of spatio-temporal points in a trajectory while preserving its essential shape and features.

**Data:** the trajectory data consisting of spatio-temporal points is {[(0.2516, 0.772114, 0.576668), (0.2561, 0.772114, 0.576674), (0.2606, 0.772114, 0.576679), (0.2651, 0.772140, 0.576860), (0.2651, 0.772140, 0.576860), (0.2696, 0.772140, 0.576809), (0.2741, 0.772135, 0.576792), (0.2786, 0.772135, 0.576713)]}.

**Information:** none.

**Format:** the output should be the trajectory data.

---

*Output*

**Model response:** the result of trajectory simplification is [(0.2516, 0.772114, 0.576668), (0.2606, 0.772114, 0.576679), (0.2651, 0.772140, 0.576860), (0.2651, 0.772140, 0.576860), (0.2696, 0.772140, 0.576809), (0.2741, 0.772135, 0.576792)].

---

Figure 10: The example of local trajectory simplification

---

**Cross-Client Trajectory Imputation**

---

*Processing*

**Raw Data 1:** [(1201955434, 116.49625, 39.9146), (1201956033, 116.50962, 39.91071), (1201956633, 116.52231, 39.91588), ()]

**Normalized Data 1:** [(0.1886, 0.672904, 0.612130), (0.1931, 0.679691, 0.609936), (0.1976, 0.686134, 0.612852), ()]

**Raw Data 2:** [(1201957833, 116.59512, 39.90798), (1201958433, 116.61153, 39.88277), (1201959033, 116.65522, 39.8622)]

**Normalized Data 2:** [(0.2066, 0.723098, 0.608396), (0.2111, 0.731429, 0.594173), (0.2156, 0.753610, 0.582569)]

---

*Prompt*

**Task:** it is a trajectory imputation task. It aims to reconstruct a complete trajectory by predicting or estimating the missing points based on available spatio-temporal data. This often occurs when GPS signals are lost or data collection is interrupted.

**Data:** the trajectory data consisting of spatio-temporal points is [p1, p2, p3, (), p4, p5, p6].

**Information:** none.

**Format:** the output should be the trajectory data.

---

*Output*

**Model response:** the result of trajectory imputation is [p1, p2, p3, p4, p5, p6, p7].

---

Figure 11: The example of cross-client trajectory imputation

Table 5: The statistics of dataset

| Dataset | # trajectories | # points | Quality Issue | Task |
|---|---|---|---|---|
| GeoLife | 182 | 24,876,978 | Positional inaccuracies, data noise, and lower precision | AD, TI, MM, TUL, TMI, TSim, and TR |
| Porto | 442 | 83,409,386 | Anomalies and missing data | AD and TI |
| T-Drive | 10,336 | 17,662,984 | Noisy and incomplete points | NF, TR, SPD, and TSim |
| Tencent | 40,966 | 1,610,216 | Inaccurate points | MM |
| Gowalla | 107,092 | 6,442,890 | Sparse and non-continuous data | TUL |
| SHL | 3 | 109,390 | Duplicate records | TSeg and TMI |

# D    EXPERIMENTAL DETAILS

## D.1    DATASETS

We evaluate the proposed FedTDP framework using 6 datasets, including GeoLife (Zheng et al., 2010), Porto (2015), T-Drive (Yuan et al., 2010), Tencent (Liu et al., 2024b), Gowalla (Cho et al., 2011), and SHL (2017), with their statistics shown in Table 5, as detailed below.

- **GeoLife.** It collected 182 trajectories with 24,876,978 spatio-temporal points, used for the training tasks, including Anomaly Detection (AD), Trajectory Imputation (TI), Map Matching (MM), Trajectory-User Link (TUL), Travel Mode Identification (TMI), Trajectory Simplification (TSim), and Trajectory Recovery (TR) tasks. It contains various quality issues such as positional inaccuracies, data noise, and lower precision due to irregular sampling intervals and sensor limitations, which make it suitable for various trajectory data preparation tasks.

- **Porto.** It collected 442 trajectories with 83,409,386 spatio-temporal points, which contains quality issues such as anomalies and missing data, used for AD and TI testing tasks.

- **T-Drive.** It collected 10,336 trajectories with 17,662,984 spatio-temporal points, which contains quality issues such as noisy and incomplete points, used for NF, TR, SPD, and TSim testing tasks.

- **Tencent.** It collected 40,966 trajectories with 1,610,216 spatio-temporal points, which contains quality issues such as inaccurate points due to the low sampling rate, used for the MM testing task.

- **Gowalla.** It collected 107,092 trajectories with 6,442,890 spatio-temporal points, which contains quality issues such as sparse and non-continuous data, used for the TUL testing task.

- **SHL.** It collected 3 trajectories with 109,390 spatio-temporal points, which contain quality issues such as duplicate records, used for TSeg and TMI testing tasks.

## D.2    BASELINES

We compare the proposed FedTDP framework with state-of-the-art baselines, as shown in Table 6.

First, we compare FedTDP with various Trajectory Data Preparation (TDP) methods in a single TDP task (referred to S-TDP), including ATROM (Gao et al., 2023) for the AD task, Kamel (Musleh & Mokbel, 2023) for the TI task, GraphMM (Liu et al., 2024b) for the MM task, AttnTUL (Chen et al., 2024b) for the TUL task, Estimator (Hu et al., 2024a) for the TMI task, S3 (Fang et al., 2023) for the TSeg task, and LightTR (Liu et al., 2024c) for the TR task, as detailed below.

- **ATROM.** It solves the anomaly detection task in open-world scenarios and introduces a new probabilistic metric learning model.

- **Kamel.** It proposes a scalable system that inserts additional real trajectory points to improve the accuracy of the trajectory imputation task.

- **GraphMM.** It develops the graphical nature of the map matching task to exploit the road network and trajectory graphical topology.

- **AttnTUL.** It proposes a hierarchical trajectory attention neural network for co-encoding local trajectory transition patterns and global spatial dependencies to solve the trajectory-user link task.

- **Estimator.** It partitions the entire traffic space into disjoint spatial regions based on the traffic conditions for the travel mode identification task.

Table 6: The compared baselines

| Category | Method | Task | Year |
|---|---|---|---|
| **S-TDP** | ATROM | Anomaly Detection | 2023 |
| | Kamel | Trajectory Imputation | 2023 |
| | GraphMM | Map Matching | 2024 |
| | AttnTUL | Trajectory-User Linking | 2024 |
| | Estimator | Travel Mode Identification | 2024 |
| | S3 | Trajectory Simplification | 2023 |
| | LightTR | Trajectory Recovery | 2024 |
| **Large Language Models for Table Data Preparation** | FM4DP | All tasks for evaluation | 2022 |
| | MELD | | 2024 |
| | TableGPT | | 2024 |
| **Large Language Models for Spatio-Temporal Data Analysis** | PromptGAT | | 2024 |
| | UniST | | 2024 |
| | UrbanGPT | | 2024 |

- **S3.** It presents a lightweight trajectory segmentation task framework to augment the trajectory representation paradigm with geo-semantics.

- **LightTR.** It develops a local trajectory embedding module that provides higher computational efficiency for the trajectory recovery task.

Besides, we compare FedTDP with three methods using Large Language Models (LLMs) for table data preparation, including FM4DP (Narayan et al., 2022), MELD (Yan et al., 2024), and TableGPT (Li et al., 2024a), as detailed below.

- **FM4DP.** It helps LLMs understand table DP tasks, which uses 5 data cleaning and integration table DP tasks as prompt tasks and evaluates the performance of LLMs on these tasks.

- **MELD.** It employs the mixture-of-experts architecture to support the merging and augmentation of experts trained on the domain-specific experts trained on limited annotated examples.

- **TableGPT.** It proposes a table-tuning paradigm using various table tasks synthesized from real tables as the training data to help LLMs understand the table data and perform table tasks.

Moreover, we compare FedTDP with three LLM-based models for spatio-temporal data analysis, including PromptGAT (Da et al., 2024), UniST (Yuan et al., 2024), and UrbanGPT (Li et al., 2024b), as detailed below.

- **PromptGAT.** It uses the LLM to analyze system dynamics, leveraging the context and spatio-temporal data to understand how weather, traffic, and road conditions affect traffic dynamics.

- **UniST.** It proposes a general-purpose model, which is designed for urban spatio-temporal prediction in various urban scenarios to capture the complex spatio-temporal relationship.

- **UrbanGPT.** It integrates a spatio-temporal dependency encoder with a command adjustment paradigm, which enables LLMs to understand complex spatio-temporal interdependencies.

D.3 EVALUATION METRICS AND IMPLEMENTATIONS

For evaluation, we use Synchronized Euclidean Distance (SED) for the trajectory simplification task and $F_1$ scores for other tasks, where lower SED and higher $F_1$ indicate better performance. Model efficiency is further assessed by runtime and communication size.

All baselines are run under their optimal settings. To ensure fairness in F-TDP, we extend baselines with differential privacy (Dwork et al., 2006), where clients perturb local trajectory data before transmitting to the server, since FedTDP inherently preserves privacy through its TPA module. All experiments are conducted in a federation of 9 nodes (1 server and 8 clients), each equipped with two Intel Xeon CPU E5-2650 12-core processors, two GeForce RTX 3090 GPUs, and 100 MB/s internet bandwidth. To simulate the real-world F-TDP scenario, we partition each dataset according to geographic regions. Specifically, we divide the geographic area covered by a dataset into 8 non-overlapping regions, each assigned to a distinct client. A complete trajectory that traverses multiple regions is segmented at the region boundaries, and each resulting sub-trajectory is stored locally by the corresponding client.

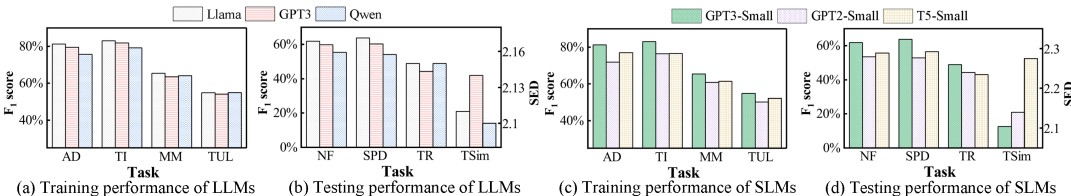

Figure 12: Large language model and small language model base study

## D.4 MODEL BASE STUDY

To evaluate the impact of various model bases on the proposed FedTDP framework, we choose widely used model bases for the LLM (Llama-8B (Touvron et al., 2023), GPT3-7B (Brown et al., 2020), and Qwen-7B (Bai et al., 2023)) and SLM (GPT3-Small-125M (Brown et al., 2020), GPT2-Small-137M (Radford et al., 2019), and T5-Small-60M (Raffel et al., 2020)). Besides, to evaluate the impact of different LLMs on FedTDP, we use GPT3-Small as the client's SLM while we use Llama as the server's LLM to evaluate the impact of different SLMs on FedTDP. The results are shown in Fig. 12. As observed, Llama achieves optimal performance in most TPD tasks for the server's LLM, followed by GPT3 and then Qwen. In contrast, GPT3-Small demonstrates the best performance for the client's SLM, succeeded by T5-Small and then GPT2-Small. Consequently, we adopt Llama and GPT3-Small as the default server's LLM and client's SLM in other experiments, respectively.

## D.5 PARAMETER SENSITIVITY STUDY

We evaluate the effects of hyperparameters of the proposed FedTDP framework (i.e., the training layers ratio $m$ of LoRA sparse-tuning in the TKE module), as shown in Fig. 13, where we change the $m$ from 25% to 100%. We can observe that as $m$ increases, the performance of FedTDP improves slightly. However, this improvement comes at the cost of increased training time and communication size, as the number of parameters that need to be trained and transmitted also rises when $m$ is increased. Therefore, the suggested value of $m$ is 25% or less, as long as the model performance with the value of $m$ is acceptable.

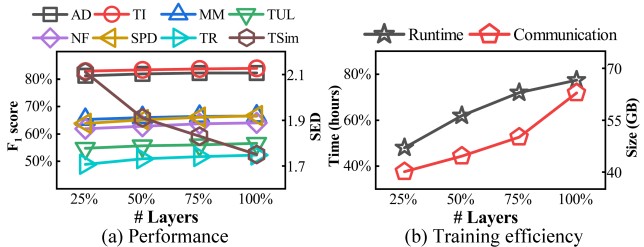

Figure 13: Parameter sensitive study

## E LIMITATION AND FUTURE STUDY

Despite its promising results, this work has certain limitations.

First, while FedTDP excels at preparing trajectory data for downstream tasks, it is not designed as a direct analysis engine for tasks like clustering, pattern mining, or long-term forecasting. Specifically, many downstream analysis tasks are highly sensitive to the quality of input data. FedTDP's strength in noise filtering and imputation directly enhances the signal-to-noise ratio for these tasks. However, a model trained on FedTDP-processed data may still fail if the semantic structure of the data (e.g., complex movement patterns, high-level behavioral semantics) is inherently beyond the representation capacity of the underlying LLM/SLM. For instance, while FedTDP can perfectly map a GPS point to a road segment, it does not inherently learn the motivation behind a detour or the cultural context of a stay point. Extending FedTDP to support these tasks would require fundamentally rethinking its objective: shifting from a data preprocessor to a semantic interpreter. This would necessitate integrating task-specific knowledge graphs, incorporating higher-order temporal dependencies, and potentially moving away from the pure prompt-based paradigm to a more end-to-end generative architecture, which would dramatically increase complexity and computational cost. We view this as

a critical, non-trivial research direction, and our current work establishes a necessary, high-quality data foundation upon which such advanced analysis models can be built.

The use of LLMs as the core reasoning engine for TDP tasks introduces a significant black box challenge. While our TKE module provides a structured prompt, the internal decision-making process within the LLM/SLM remains opaque. The current framework outputs a cleaned trajectory or a classification result, but it provides no explanation for why a point was flagged as an anomaly, how a missing segment was inferred, or which spatio-temporal features were decisive in a map-matching decision. This lack of interpretability has three critical implications:

- Trust and Adoption: Domain experts (e.g., urban planners, traffic engineers) are unlikely to trust or deploy a system whose reasoning they cannot audit. A black box TDP system is unusable in safety-critical applications.
- Debugging and Error Analysis: When the system makes an error (e.g., misclassifying a noisy point as a valid stay point), it is currently impossible to diagnose whether the fault lies in the TPA encoding, the prompt design, the LLM's internal knowledge, or the distillation process.
- Bias and Fairness: LLMs can inadvertently encode biases present in their pre-training data (e.g., favoring car travel over walking in dense urban areas). Without interpretability, we cannot detect or mitigate such biases in the TDP output.

Future work must integrate techniques like attention visualization, counterfactual explanations, or LLM-as-a-judge frameworks to provide post-hoc justifications for FedTDP's decisions, transforming it from a purely functional tool into a transparent, accountable system.

## F  LLM USAGE STATEMENT

In the preparation of this manuscript, Large Language Models (LLMs) were used solely as a general-purpose writing assistance tool to improve the clarity, grammar, and fluency of the text. The core research ideas, experimental design, data analysis, and all technical content were conceived and developed entirely by the human authors. No LLM was involved in generating novel ideas, interpreting results, or producing scientific claims. The authors take full responsibility for the accuracy and integrity of all content presented in this paper.

