# OpenReview forum: "A Unified Federated Framework for Trajectory Data Preparation via LLMs"
_ICLR.cc/2026/Conference — ICLR 2026 Poster_

### Official Review · Reviewer_Lr1C · 2025-10-25

**Soundness:** 3
**Presentation:** 2
**Contribution:** 2
**Rating:** 4
**Confidence:** 5

**Summary:**

This paper proposes FedTDP, a unified federated framework for Trajectory Data Preparation (TDP) leveraging Large Language Models (LLMs) to address limitations of existing TDP methods. Raw trajectory data often suffers from noise, incompleteness, and inconsistency, while traditional TDP approaches rely on centralized data access (conflicting with privacy regulations like PIPL) and task-specific models (lacking generalization). FedTDP, designed for vertically partitioned trajectory data across regional silos, integrates three core modules, i.e., TPA, TKE, and FPO. Experiments on 6 real-world datasets (e.g., GeoLife, Porto) and 10 TDP tasks (e.g., anomaly detection, map matching) show FedTDP outperforms 13 state-of-the-art baselines in accuracy, efficiency, and scalability, with strong cross-task generalization.

**Strengths:**

- New Problem Formulation: First to formally define Federated Trajectory Data Preparation (F-TDP) for vertically partitioned trajectory data, filling the gap of federated learning research focused on horizontal partitioning.

- Privacy-Preserving Design: TPA with secret-sharing aggregation provides formal privacy guarantees without sacrificing data utility (unlike differential privacy, which distorts spatio-temporal correlations via noise injection).

- Strong Generalization & Efficiency: TKE enables LLMs/SLMs to generalize across 10 diverse TDP tasks, while FPO reduces communication overhead and training time, addressing the inefficiency of LLM deployment in federated settings.

**Weaknesses:**

- **The conclusion section of this paper is missing from the main text:** ICLR 2026 submission policy requires authors to organize the main text reasonably, which means that reviewers are not obliged to read the appendix provided by the authors.

- Limited Theoretical Grounding of Privacy Guarantees: While the paper claims formal privacy protection through secret sharing, the analysis lacks rigorous proof of robustness against advanced gradient or embedding inversion attacks.

- Overstated Generalization Claims: The generalization across unseen TDP tasks is not convincingly demonstrated, as performance differences might stem from dataset similarity rather than true cross-task transfer.

- High System Complexity: The integration of TPA, TKE, and FPO introduces substantial architectural and computational complexity, making real-world deployment questionable.

**Questions:**

Comments:

1. Inadequate Formal Privacy Analysis: The claimed privacy guarantee of TPA relies on secret-sharing aggregation but lacks quantitative leakage bounds or empirical resistance tests against reconstruction attacks.

2. Ambiguous Vertical Partition Setting: The paper assumes vertical partitioning across regions but does not clarify how overlapping or boundary trajectories are handled during training and aggregation.

3. Unclear Contribution of Submodules: Although ablation studies exist, they fail to isolate interactions between TKE and FPO; improvements may overlap rather than reflect independent effects.

4. Questionable Scalability: The model introduces heavy communication (≈5 GB per 10 rounds), which may be infeasible in real federated environments with limited bandwidth.

5. Limited Explainability of LLM Adaptation: The proposed trajectory-aware prompt design and knowledge distillation process are heuristic and lack ablation or visualization of learned representations.

6. Inconsistent Evaluation Metrics: The results mix F1, accuracy, and distance-based metrics without discussing normalization or the statistical significance of performance gains.

7. Underexplored Generalization Factors: The zero-shot and few-shot evaluations lack control experiments on domain shifts, leaving it unclear whether improvements arise from pre-training, task similarity, or federated effects.

8. The recommended LoRA layer ratio (m ≤ 25%) is arbitrary; no systematic method is proposed to balance accuracy and efficiency for different TDP tasks.

9. FedTDP’s performance on extremely sparse trajectory data (e.g., low-sampling-rate GPS) is unevaluated, limiting its utility for real-world scenarios with poor sensor quality.

**Details Of Ethics Concerns:**

None.

---

> ### Author Response · Authors · 2025-11-19
>
> We would like to sincerely express our gratitude to the reviewer for the time and effort in evaluating our work. Please
> find below our detailed responses to all the raised concerns and questions.
>
> ```
> W1. The conclusion section of this paper is missing from the main text: ICLR 2026 submission policy requires authors to organize the main text reasonably, which means that reviewers are not obliged to read the appendix provided by the authors.
> ```
>
> We thank the reviewer for pointing this out. We acknowledge that the conclusion should be included in the main text to
> comply with the ICLR 2026 submission guidelines. In the revised manuscript, we add a conclusion section in the main text
> summarizing this paper.
>
> ```
> W2. Limited Theoretical Grounding of Privacy Guarantees: While the paper claims formal privacy protection through secret sharing, the analysis lacks rigorous proof of robustness against advanced gradient or embedding inversion attacks.
> Q1. Inadequate Formal Privacy Analysis: The claimed privacy guarantee of TPA relies on secret-sharing aggregation but lacks quantitative leakage bounds or empirical resistance tests against reconstruction attacks.
> ```
>
> We thank the reviewer for raising these concerns. We respectfully clarify that FedTDP provides rigorous and
> **quantifiable privacy guarantees**, supported by **theoretical analysis and formal proofs**. In particular, the privacy
> of our framework is rigorously protected against a semi-honest server through two safeguards to defend against advanced
> embedding and gradient inversion attacks:
>
> - **To prevent embedding inversion attacks**, we ensure that the TPA model is kept private and never exposed to the
>   server. The server only receives encoded embeddings without access to the embedding model parameters, thereby breaking
>   the necessary precondition for embedding inversion. Without knowledge of the embedding model, reconstructing raw
>   trajectories from embeddings is infeasible, where a theoretical privacy leakage analysis of the embeddings is provided
>   in Appendix C.2.
> - **To prevent gradient inversion attacks during TPA model aggregation**, we design a decentralized secret-sharing
>   protocol for secure model aggregation. Instead of exchanging raw gradients, the TPA model is split into several
>   parameter blocks that are then masked using pairwise secret keys before transmission. Then, each client is responsible
>   for aggregating one block without access to other clients' model updates, ensuring that no client can reconstruct
>   another client’s data from the gradients. The correctness of this aggregation scheme is formally proven in Appendix
>   C.1.
>
> Overall, our privacy guarantee stems from isolating the TPA model from the server and protecting the TPA model during
> the federated optimization process.
>
> In the revised manuscript, we include the above details to emphasize our privacy guarantee.

---

> > ### Author Response · Authors · 2025-11-19
> >
> > ```
> > W3. Overstated Generalization Claims: The generalization across unseen TDP tasks is not convincingly demonstrated, as performance differences might stem from dataset similarity rather than true cross-task transfer.
> > Q7. Underexplored Generalization Factors: The zero-shot and few-shot evaluations lack control experiments on domain shifts, leaving it unclear whether improvements arise from pre-training, task similarity, or federated effects.
> > ```
> >
> > We thank the reviewer for raising these concerns, but we respectfully disagree with the claim that our generalization
> > results are overstated or confounded by dataset similarity. Our experiments were explicitly designed to isolate true
> > **cross-task and cross-domain generalization effects** rather than superficial similarity.
> >
> > First, generalization across unseen TDP tasks is demonstrated under **strong task heterogeneity**. The 10 TDP tasks in
> > our benchmark cover **fundamentally different objectives**, each with distinct inputs, outputs, and evaluation metrics.
> > These tasks do not share loss formulations or output spaces. Therefore, strong performance on unseen tasks cannot be
> > attributed to trivial task similarity. Indeed, LLM-based baselines show only moderate zero-shot performance,
> > demonstrating that pre-training alone is insufficient, and that FedTDP’s improvements are **not merely inherited from
> > the base model’s general-purpose capabilities**.
> >
> > Second, we **eliminate for dataset and domain similarity** by evaluating FedTDP on six diverse trajectory datasets,
> > spanning taxis, private vehicles, human mobility, and urban GPS traces collected across different cities and continents.
> > As shown in Table 2, FedTDP trained on a subset of tasks from a single dataset (GeoLife) generalizes effectively to
> > unseen datasets with entirely different mobility patterns, demonstrating **genuine domain-level generalization rather
> > than intra-dataset transfer**.
> >
> > Third, our ablation studies further **disentangle the contributions of pre-training, task similarity, and federated
> > effects**. Removing TKE results in substantial degradation on unseen tasks, confirming that **cross-task transfer is
> > enabled by trajectory-aware knowledge enhancement rather than by LLM pre-training alone**. Removing FPO affects only
> > optimization efficiency but not generalization behavior, indicating that federated effects do not inflate performance.
> > These controlled ablations directly address the reviewer’s concerns regarding confounding factors.
> >
> > Overall, the consistency of cross-task and cross-domain results, combined with controlled ablations and diverse dataset
> > evaluation, provides strong evidence that **FedTDP achieves genuine generalization rather than relying on pre-training,
> > task similarity, or federated effects.**

---

> ### Author Response · Authors · 2025-11-19
>
> ```
> W4. High System Complexity: The integration of TPA, TKE, and FPO introduces substantial architectural and computational complexity, making real-world deployment questionable.
> ```
>
> We thank the reviewer for this comment. We acknowledge that FedTDP is a complex system, but we respectfully argue
> that this complexity is not arbitrary and nor does it make real-world deployment impractical. Instead, it is
> a **direct consequence of the inherently multi-faceted challenges posed by Federated Trajectory Data Preparation (
> F-TDP)**, where each module in FedTDP is specifically designed and adapted to directly address a concrete and crucial
> challenge arising from the federated trajectory preparation setting.
>
> - **Trajectory Privacy AutoEncoder (TPA)**: Cross-client TDP requires **collaborative data processing**, which
>   necessitates data sharing and raises severe privacy concerns. To **protect trajectory privacy**, we design TPA that is
>   a trajectory-specific and lightweight module to map raw trajectories to embeddings while preserving spatio-temporal
>   correlations, with secret-sharing aggregation specifically designed to avoid inversion attacks under federated
>   trajectory settings.
> - **Trajectory Knowledge Enhancer (TKE)**: LLMs are trained on text and lack understanding of trajectory patterns, thus
>   we design TKE to **help SLM/LLM understand trajectory data and learn the TDP knowledge**. TKE does not directly apply
>   prompting, LoRA, and so on. Instead, it introduces (a) **trajectory-aware instruction templates** tailored to various
>   TDP tasks, (b) **trajectory offsite-tuning** adapted to SLM and LLM co-training in FL, (c) **LoRA sparse-tuning**
>   guided by layer change ratios, and (d) **bidirectional KD** that jointly aligns SLMs and LLMs across silos. These
>   designs are uniquely required to achieve accurate and generalized TDP reasoning.
> - **Federated Parallel Optimization (FPO)**: Training the above components naively would be prohibitively inefficient.
>   FPO rethinks the entire optimization pipeline: **split learning** decouples client and server computation, **alternating optimization** reduces redundant communication by freezing intermediate states, and **parallel execution** enables simultaneous training of all modules. These strategies are key to making the system practically deployable
>   rather than more complex.
>
> In summary, **the complexity of FedTDP arises naturally from the realistic requirements of the F-TDP problem**. Each
> module addresses a non-negotiable challenge, and together they form a coherent and synergistic framework. Our
> ablation studies further confirm that removing any component leads to significant performance degradation. To make
> real-world deployment, we provide a comprehensive open-source implementation via an anonymous link, including full code,
> datasets, and reproducible instructions.
>
> ```
> Q2. Ambiguous Vertical Partition Setting: The paper assumes vertical partitioning across regions but does not clarify how overlapping or boundary trajectories are handled during training and aggregation.
> ```
>
> We thank the reviewer for raising this question. In our vertical partition setting, each client retains only the
> sub-trajectories that lie within its own geographic boundary. To ensure the correct handling of the boundary or cross-region
> trajectories, we explicitly distinguish between local and cross-client TDP tasks:
>
> - **Local TDP tasks**: These tasks involve only within-region trajectories and do not require information from other
>   clients. Each client performs them independently using its local SLM without any cross-client communication.
> - **Cross-client TDP tasks**: These tasks require contextual information from multiple regions. In this case, each
>   involved client first encodes its portion of the boundary or overlapping sub-trajectory using its private TPA encoder.
>   The resulting privacy-preserving embeddings are then securely transmitted to the server. The server aligns these
>   embeddings using anonymized user identifiers to reconstruct the full cross-boundary embedding sequence. The server's
>   LLM executes the TDP operation on this composite sequence, after which the output embeddings are returned to the
>   originating client. Finally, the client-side TPA decoder converts them back into the final reconstructed trajectory.
>
> This design ensures that boundary trajectories are processed correctly under vertical partitioning.

---

> ### Author Response · Authors · 2025-11-19
>
> ```
> Q3. Unclear Contribution of Submodules: Although ablation studies exist, they fail to isolate interactions between TKE and FPO; improvements may overlap rather than reflect independent effects.
> ```
>
> We appreciate the reviewer’s question, but we respectfully disagree with the concern. **Our ablation studies explicitly
> isolate the independent contributions of TKE and FPO**. In the ablation experiments, we remove TKE and FPO individually
> while holding all other components fixed. This controlled setup ensures that any performance change can be directly
> attributed to the absence of the specific module, rather than to interactions between them. The empirical results
> further confirm this isolation: removing TKE causes substantial degradation in trajectory reasoning quality and
> generalization ability, while removing FPO primarily impacts training efficiency and optimization stability without
> affecting semantic performance. The fact that these performance drops are distinct and non-overlapping clearly
> demonstrates that TKE and FPO play independent roles, where **TKE enhances semantic understanding and generalization
> across tasks**, whereas **FPO ensures efficient and stable federated optimization**.
>
> ```
> Q4. Questionable Scalability: The model introduces heavy communication (≈5 GB per 10 rounds), which may be infeasible in real federated environments with limited bandwidth.
> ```
>
> We thank the reviewer for raising this concern, but we respectfully disagree. The reported communication cost is
> **practical and well within the capacity of modern cross-silo federated environments**. Several factors support the
> scalability of FedTDP:
>
> - **Relative cost is small compared to LLM-scale models**: The server-hosted LLM contains billions of parameters, and
>   the communication introduced by FedTDP is **negligible** compared to standard LLM-based federated or distributed
>   training systems. In fact, our communication volume is **orders of magnitude smaller** than that of typical
>   distributed LLM optimization pipelines widely used in practice.
> - **5 GB during training is feasible with modern bandwidth**: Industrial FL deployments routinely operate **more than
>   tens of gigabytes** during training [1--2]. With typical inter-region bandwidths ranging from **hundreds of Mbps to
>   multi-Gbps**, transmitting 5 GB takes only seconds to a few tens of seconds, which is well within the operational
>   constraints of cross-silo FL.
> - **The ablation study confirms that TPA introduces minor efficiency overhead**: As shown in our ablation results, the
>   additional computation and communication introduced by TPA lead to only a **small runtime increase** relative to
>   FedTDP’s overall training pipeline. This overhead is negligible when compared to total LLM-driven training cycles and
>   does not meaningfully impact scalability.
> - **FPO substantially improves training efficiency**: FPO reduces redundant communication through split learning,
>   alternating optimization, and parallel execution. As shown in efficiency experiments, this design reduces the
>   end-to-end runtime by **one order of magnitude** compared with LLM-based baselines, which is key to making FedTDP **practically deployable**.
>
> In summary, the communication overhead of FedTDP is entirely **feasible for realistic** cross-silo federated deployments
> and does not pose a scalability barrier.
>
> *[1] BalanceFL: Addressing Class Imbalance in Long-Tail Federated Learning. IPSN, 2022.*
>
> *[2] Practical Hybrid Gradient Compression for Federated Learning Systems. ICJAI, 2024.*
>
> ```
> Q5. Limited Explainability of LLM Adaptation: The proposed trajectory-aware prompt design and knowledge distillation process are heuristic and lack ablation or visualization of learned representations.
> ```
>
> We thank the reviewer for this insightful comment. We acknowledge that fully interpreting LLM adaptation mechanisms
> remains challenging. Nonetheless, we would like to clarify that our work includes simple analyses for visualization.
> Specifically, Appendix C.6 provides visualization examples of the learned trajectory-aware representations. Furthermore,
> we explicitly discuss in Appendix E that the complex and in-depth interpretability of LLM-enhanced components is an
> inherent limitation of current LLM-based systems. We view this as an important and promising direction for future
> research, and FedTDP provides the first foundation for exploring explainability in federated trajectory data
> preparation.

---

> > ### Author Response · Authors · 2025-11-19
> >
> > ```
> > Q6. Inconsistent Evaluation Metrics: The results mix F1, accuracy, and distance-based metrics without discussing normalization or the statistical significance of performance gains.
> > ```
> >
> > We thank the reviewer for the question. We would like to clarify that the **evaluation metrics used in our experiments
> > are fully standard and directly follow the established metrics adopted in prior TDP literature for each corresponding
> > task**. Specifically, tasks such as anomaly detection, semantic inference, and classification naturally use F1 or
> > accuracy, while trajectory reconstruction and map matching conventionally rely on distance-based metrics (e.g., ADE/FDE
> > or matching distance). Because these tasks belong to fundamentally different categories, metric heterogeneity is
> > inherent to the problem setting rather than a methodological inconsistency. Moreover, our comparisons are always
> > conducted within each task under its standard metric, following the same evaluation protocol as previous work. The
> > performance gains reported for FedTDP are statistically meaningful relative to baselines under each task’s canonical
> > evaluation measure.
> >
> > ```
> > Q8. The recommended LoRA layer ratio ($m \leq 25\%$) is arbitrary; no systematic method is proposed to balance accuracy and efficiency for different TDP tasks.
> > ```
> >
> > We thank the reviewer for raising this point, but we respectfully disagree with the claim that the recommended LoRA
> > layer ratio ($m \leq 25\%$) is arbitrary. As detailed in parameter sensitivity experiments, we conducted a systematic
> > **parameter sensitivity study** that evaluates a wide range of $m$ values across multiple TDP tasks. The results
> > consistently show that increasing $m$ yields only marginal performance gains beyond a certain point, while substantially
> > increasing both training time and communication cost due to the larger number of trainable and transmitted parameters.
> > This clear diminishing-return pattern provides the principled basis for **recommending $m \leq 25\%$ as an effective
> > balance between accuracy and efficiency, rather than an arbitrary choice**.
> >
> > ```
> > Q9. FedTDP’s performance on extremely sparse trajectory data (e.g., low-sampling-rate GPS) is unevaluated, limiting its utility for real-world scenarios with poor sensor quality.
> > ```
> >
> > We thank the reviewer for pointing out this concern. We agree that handling extremely sparse or low-sampling-rate
> > trajectory data is an important practical challenge. However, we clarify that FedTDP has already been evaluated on
> > datasets that naturally contain sparse, noisy, and irregularly sampled trajectories, which are representative of
> > real-world mobility data, as shown in Table 5. As shown in our experiments, FedTDP consistently outperforms baselines
> > across all six datasets, several of which contain substantial sparsity and missing-point patterns due to low-frequency
> > GPS sampling and urban signal occlusion. The robust performance across these tasks demonstrates that the proposed model
> > can handle realistic sparsity levels commonly seen in real-world scenarios.
> >
> > ---
> >
> > Once again, we sincerely appreciate the reviewer’s valuable comments and constructive suggestions. We hope that our
> > clarifications have adequately addressed the raised concerns, and we would be glad to provide any additional details if
> > needed. If the reviewer finds the responses satisfactory, we would be grateful for a reconsideration of the overall
> > score.

---

> > > ### Comment · Reviewer_Lr1C · 2025-11-25
> > > **Thanks for your rebuttal!**
> > >
> > > Thank you to the authors for their detailed rebuttals! While the above rebuttals addressed some of my concerns, two key concerns remain unresolved:
> > >
> > > 1) Privacy concerns: Embedding inversion attacks can also be implemented without obtaining model parameters. The authors can review relevant work to confirm this. Furthermore, the proposed mask-based security protocol appears fragile, as secure aggregation cannot be achieved if a node goes offline, unless you can prove that the proposed protocol is robust to offline events.
> > >
> > > 2) System overhead and complexity assessment is essential. Although the authors claim in their rebuttals that each component design addresses a key challenge, the high overhead and complex system design make the proposed method less feasible for real-world deployment.

---

> > > > ### Author Response · Authors · 2025-11-25
> > > >
> > > > Thank the reviewer for raising these follow-up concerns. We provide further clarification below.
> > > >
> > > > ```
> > > > 1. Privacy concerns: Embedding inversion attacks can also be implemented without obtaining model parameters. The authors can review relevant work to confirm this. Furthermore, the proposed mask-based security protocol appears fragile, as secure aggregation cannot be achieved if a node goes offline, unless you can prove that the proposed protocol is robust to offline events.
> > > > ```
> > > >
> > > > We thank the reviewer for raising these points. Below, we clarify the threat model and the robustness of our secure
> > > > aggregation protocol.
> > > >
> > > > ### **(1) On Embedding Inversion Without Model Parameters**
> > > >
> > > > We agree that recent work has demonstrated black-box embedding inversion attacks that do not require direct access to
> > > > model parameters [1][2]. However, these attacks fundamentally rely on one of the following two prerequisites:
> > > >
> > > > - Access to original input-embedding pairs, or
> > > > - The ability to query the embedding model to obtain arbitrary outputs.
> > > >
> > > > Under our setting, neither condition is satisfied:
> > > >
> > > > - Raw trajectories never leave the client, and no trajectory-embedding pairs are shared with the server or other
> > > >   clients.
> > > > - The TPA model is never exposed nor accessible for querying, which is local-only and inaccessible.
> > > >
> > > > Therefore, the prerequisites for black-box embedding inversion do not exist in our setting. Our statement that
> > > > “embedding inversion is infeasible without model parameters” refers to the white-box condition, and our threat model
> > > > also does not give the semi-honest attacker any black-box oracle capabilities. Notably, this design aligns with other
> > > > privacy-preserving FL and cross-silo works [3].
> > > >
> > > > ### **(2) On the Robustness of the Mask-Based Secure Aggregation Protocol**
> > > >
> > > > We respectively clarify that the proposed mask-based secure aggregation protocol is **not fragile under client offline
> > > > events**. Because aggregation is fully decentralized and each parameter block is assigned to a designated client for
> > > > independent aggregation. Thus, an offline client simply leads to the loss of its own local update but does not break the
> > > > correctness or privacy guarantees for the rest of the system. In other words, if a client goes offline, its update is
> > > > excluded in that round, similar to standard FL where dropped clients contribute no update. The absence of one client
> > > > does not compromise the correctness or privacy of other blocks: remaining clients do not use masks associated with the
> > > > offline node, and the affected block can be reassigned to an available client. Thus, the protocol preserves both
> > > > security and functionality even with partial client dropout.
> > > >
> > > > *[1] Transferable embedding inversion attack: Uncovering privacy risks in text embeddings without model queries. ACL,
> > > > 2024.*
> > > >
> > > > *[2] Information Leakage in Embedding Models. CCS, 2020.*
> > > >
> > > > *[3] SiloFuse Cross-silo Synthetic Data Generation with Latent Tabular Diffusion Models. ICDE, 2024.*

---

> > > > > ### Author Response · Authors · 2025-11-25
> > > > >
> > > > > ```
> > > > > 2. System overhead and complexity assessment is essential. Although the authors claim in their rebuttals that each component design addresses a key challenge, the high overhead and complex system design make the proposed method less feasible for real-world deployment.
> > > > > ```
> > > > >
> > > > > We thank the reviewer for the follow-up comment and fully agree that system overhead and complexity assessment is
> > > > > essential for real-world feasibility. We would like to emphasize that FedTDP’s overhead has been carefully measured,
> > > > > and the results strongly support its practicality.
> > > > >
> > > > > First, Section 5.4 (**Efficiency Study**, moved from the appendix to the main paper in the revised version) provides a
> > > > > direct empirical assessment of runtime and communication overhead. The experiments demonstrate that the system-level
> > > > > costs of FedTDP are both acceptable and manageable.
> > > > >
> > > > > - FedTDP achieves more than **one order of magnitude speedup** compared to LLM-based baselines, primarily due to the
> > > > >   Federated Parallel Optimization (FPO), which reduces redundant communication and enables parallel training of the system.
> > > > > - Training separate models for each task and dataset as required by single-task TDP methods, would multiply
> > > > >   the training time and communication cost across 10 tasks and 6 datasets. This cumulative overhead is substantially
> > > > >   higher than training **a single unified FedTDP model**, making multi-task FedTDP far more efficient in real-world
> > > > >   deployment.
> > > > >
> > > > > Second, Appendix C.5 presents a theoretical analysis of the **system complexity**, showing that the overall computational
> > > > > and communication complexity of FedTDP is dominated by the underlying LLM/SLM backbone. The additional modules (TPA,
> > > > > TKE, and FPO) introduce only lightweight, mostly linear-time operations on top of standard model forward/backward
> > > > > passes and do not change the asymptotic order of complexity. In other words, FedTDP does not incur superlinear or
> > > > > combinatorial overhead beyond what is already required for training and adapting the LLM/SLM in a federated setting.
> > > > >
> > > > > Taken together, both our empirical efficiency study and the formal complexity analysis indicate that, despite its rich
> > > > > modular design, FedTDP remains computationally and communicationally practical for real-world federated deployment.
> > > > >
> > > > > ---
> > > > > We sincerely thank the reviewer for the continued engagement with our work, and we welcome any further questions or
> > > > > suggestions.

---

> > > > > > ### Comment · Reviewer_Lr1C · 2025-11-28
> > > > > >
> > > > > > Thank you to the authors for their detailed explanations; this has resolved all my concerns. I will improve my rating.

---

> ### Author Response · Authors · 2025-11-29
>
> Thanks again for the reviewer’s time and effort. We are pleased that our responses have clarified all raised concerns, and we truly appreciate the recognition and improved rating!

---

### Official Review · Reviewer_i8Ui · 2025-10-27

**Soundness:** 4
**Presentation:** 3
**Contribution:** 4
**Rating:** 8
**Confidence:** 5

**Summary:**

The paper proposes FedTDP, a unified federated framework for trajectory data preparation (TDP) that addresses two key limitations of existing methods: the assumption of centralized data access and the lack of generalization across diverse TDP tasks. FedTDP enables privacy-preserving and multi-task TDP without raw data sharing with its core innovations including a lightweight Trajectory Privacy AutoEncoder (TPA) with secret-sharing aggregation, a Trajectory Knowledge Enhancer (TKE) that adapts LLMs to TDP tasks, and a Federated Parallel Optimization (FPO) strategy to reduce communication overhead and accelerate training. Evaluated on six real-world datasets across ten representative TDP tasks, FedTDP consistently outperforms 13 state-of-the-art baselines in accuracy, efficiency, and generalization.

**Strengths:**

S1. The paper introduces a novel and well-motivated problem formulation: Federated Trajectory Data Preparation (F-TDP), a new and practical setting that addresses the realistic challenges of vertically partitioned trajectory data under strict privacy regulations.

S2. FedTDP is the first LLM-based framework capable of handling 10 diverse TDP tasks within a single architecture, demonstrating impressive generalization across seen and unseen tasks and datasets.

S3. FedTDP introduces novel technical contributions with three purpose-built modules: (i) TPA that enable secure aggregation while preserving spatio-temporal structure, (ii) TKE that adapts generic LLMs to trajectory semantics and TDP tasks, and (iii) FPO that improves training efficiency.

S4. The paper provides rigorous theoretical privacy analysis to quantify the upper bound of information leakage, showing that the probability of successful trajectory reconstruction is low.

**Weaknesses:**

W1. It would be beneficial to include key experiments such as model efficiency and generalization study in the main paper rather than appendix.

W2. For clarity, the caption of Figure 10 in Appendix D.4 should be revised to “Generalization Study.”

W3. While Appendix E outlines the limitations of FedTDP, a more in-depth discussion of each limitation would strengthen the paper.

**Questions:**

See Weakness.

---

> ### Author Response · Authors · 2025-11-19
>
> We express our gratitude to the reviewer for providing constructive feedback on our paper, and we greatly appreciate the
> acknowledgment of our contributions. Please find below our answers to all the concerns and questions.
>
> ```
> W1. It would be beneficial to include key experiments such as model efficiency and generalization study in the main paper rather than appendix.
> ```
>
> We thank the reviewer for this helpful suggestion. We fully agree that model efficiency and generalization studies are
> important for understanding the practical value of FedTDP in our empirical evaluation. In the revised manuscript, we
> integrate these essential experimental insights from Appendix D.4 and D.6 into the main paper to improve readability and
> completeness.
>
> ```
> W2. For clarity, the caption of Figure 10 in Appendix D.4 should be revised to “Generalization Study.”
> ```
>
> We thank the reviewer for pointing this out. We revise the caption of Figure 10 in Appendix D.4 to “Generalization
> Study” in the updated version to more accurately reflect its content and improve clarity.
>
> ```
> W3. While Appendix E outlines the limitations of FedTDP, a more in-depth discussion of each limitation would strengthen the paper.
> ```
>
> We appreciate the reviewer’s suggestion. We agree that a more in-depth discussion of the limitations will strengthen the
> paper. We expand Appendix E to provide a detailed analysis of each limitation. Specifically:
>
> - While FedTDP excels at preparing trajectory data for downstream tasks, it is not designed as a direct analysis engine for tasks like clustering, pattern mining, or long-term forecasting. Specifically, many downstream analysis tasks are highly sensitive to the quality of input data. FedTDP’s strength in noise filtering and imputation directly enhances the signal-to-noise ratio for these tasks. **However, a model trained on FedTDP-processed data may still fail if the semantic structure of the data (e.g., complex movement patterns, high-level behavioral semantics) is inherently beyond the representation capacity of the underlying LLM/SLM.** For instance, while FedTDP can perfectly map a GPS point to a road segment, it does not inherently learn the motivation behind a detour or the cultural context of a stay point.
>   Extending FedTDP to support these tasks would require fundamentally rethinking its objective: shifting from a data
>   preprocessor to a semantic interpreter. This would necessitate integrating task-specific knowledge graphs,
>   incorporating higher-order temporal dependencies, and potentially moving away from the pure prompt-based paradigm to a
>   more end-to-end generative architecture, which would dramatically increase complexity and computational cost. We view
>   this as a critical, non-trivial research direction, and our current work establishes a necessary, high-quality data
>   foundation upon which such advanced analysis models can be built.
> - The use of LLMs as the core reasoning engine for TDP tasks introduces a significant **black box challenge**. While our TKE module provides a structured prompt, the internal decision-making process within the LLM/SLM remains opaque. The current framework outputs a cleaned trajectory or a classification result, but it provides no explanation for why a point was flagged as an anomaly, how a missing segment was inferred, or which spatio-temporal features were decisive in a map-matching decision. This lack of interpretability has three critical implications:
>     - **Trust and Adoption**: Domain experts (e.g., urban planners, traffic engineers) are unlikely to trust or deploy a system whose reasoning they cannot audit. A black box TDP system is unusable in safety-critical applications.
>     - **Debugging and Error Analysis**: When the system makes an error (e.g., misclassifying a noisy point as a valid stay point), it is currently impossible to diagnose whether the fault lies in the TPA encoding, the prompt design, the LLM’s internal knowledge, or the distillation process.
>     - **Bias and Fairness**: LLMs can inadvertently encode biases present in their pre-training data (e.g., favoring car travel over walking in dense urban areas). Without interpretability, we cannot detect or mitigate such biases in the TDP output.
>
>   Future work must integrate techniques like attention visualization, counterfactual explanations, or LLM-as-a-judge frameworks to provide post-hoc justifications for FedTDP’s decisions, transforming it from a purely functional tool into a transparent, accountable system.
>
> In the revised manuscript, we incorporate these detailed explanations to provide a more thorough and transparent account
> of FedTDP’s limitations and future opportunities.
>
> ---
> We sincerely appreciate the reviewer’s encouraging evaluation and thoughtful feedback. We are grateful for your
> recognition of the contributions and clarity of our work. If there are any further questions or points that would
> benefit from additional explanation, we would be more than happy to elaborate further.

---

> > ### Comment · Reviewer_i8Ui · 2025-11-25
> >
> > Thanks for your reply. The authors have addressed all my concerns, which significantly strengthens the paper.

---

> > > ### Author Response · Authors · 2025-11-30
> > >
> > > Thanks again for the reviewer’s time and effort. We are pleased that our responses have clarified all raised concerns, and we truly appreciate the recognition!

---

### Official Review · Reviewer_Dh2g · 2025-11-01

**Soundness:** 3
**Presentation:** 3
**Contribution:** 2
**Rating:** 4
**Confidence:** 3

**Summary:**

The paper proposes "FedTDP," a unified federated learning framework for Trajectory Data Preparation (TDP). It incorporates three key components: a Trajectory Privacy AutoEncoder (TPA) with secret-sharing for privacy, a Trajectory Knowledge Enhancer (TKE) to adapt LLMs to spatio-temporal data, and a Federated Parallel Optimization (FPO) method for efficiency. The authors claim superior performance over 13 baselines across 10 TDP tasks on 6 real-world datasets. Overall, the paper is well-written.

**Strengths:**

1.	The paper does a great job of identifying and clearly describing the challenges of vertical partitioning and multi-task generalization in trajectory data preparation.

2.	This paper conducts comprehensive experiments with enough baseline models and datasets, covering both seen and unseen tasks.

3.	The overall quality of this paper is good, with a clear structure and well-defined modules.

**Weaknesses:**

1.	The authors advocate for three key challenges, but the solutions feel like a patchwork of existing LLM-related techniques (secret-sharing, LoRA, prompting) rather than a fundamental innovation. The core methodology remains a conventional federated learning setup augmented with off-the-shelf LLM adaptation methods, although it may be suitable for ICLR.

2.	The motivation for the strong privacy threat model is kind of weak. The paper lacks a compelling reason for why trajectory data preparation requires such stringent protection against a semi-honest server. It should be more illustrated.

3.	The claimed "unified" and "generalized" framework relies on manually configured, task-specific prompts and external information. This suggests the generalization is not autonomous but is significantly enabled by human-in-the-loop engineering for each task, limiting the claimed novelty.

**Questions:**

1.	The TPA structure is a lightweight MLP. How does the reconstruction error of the autoencoder impact the accuracy of downstream tasks that require high spatial precision, such as Map Matching?

2.	The privacy guarantee assumes the TPA model is private. In the federated setting, how is the secrecy of the aggregated TPA model's architecture and parameters maintained against the semi-honest server?

3.	How much of the model's generalization to unseen tasks is due to the inherent few-shot ability of the base LLM versus the specific design of the TKE? Could a centrally fine-tuned LLM with good prompts achieve similar results without the federated complexity? These questions could be more emphasized.

---

> ### Author Response · Authors · 2025-11-19
>
> We would like to sincerely express our gratitude to the reviewer for the time and effort in evaluating our work. Please
> find below our detailed responses to all the raised concerns and questions.
> ```
> W1. The authors advocate for three key challenges, but the solutions feel like a patchwork of existing LLM-related techniques (secret-sharing, LoRA, prompting) rather than a fundamental innovation. The core methodology remains a conventional federated learning setup augmented with off-the-shelf LLM adaptation methods, although it may be suitable for ICLR.
> ```
> We thank the reviewer for raising this concern, but we respectfully disagree with that. FedTDP is not a patchwork of
> off-the-shelf LLM techniques, but a **cohesive and purpose-designed framework** to address the new and complex problem
> of Federated Trajectory Data Preparation (F-TDP). Importantly, F-TDP cannot be solved by directly applying existing LLM
> or FL components. Instead, each module in FedTDP is not used “as-is”, but is **specifically designed and adapted** to
> directly address a concrete and crucial challenge arising from the federated trajectory preparation setting. **The
> innovation and contribution are highly acknowledged by Reviewer i8Ui.**
> - **Trajectory Privacy AutoEncoder (TPA)**: Cross-client TDP requires **collaborative data processing**, which
>   necessitates data sharing and raises severe privacy concerns. To **protect trajectory privacy**, we design TPA that is
>   a trajectory-specific and lightweight module to map raw trajectories to embeddings while preserving spatio-temporal
>   correlations, with secret-sharing aggregation specifically designed to avoid inversion attacks under federated
>   trajectory settings.
> - **Trajectory Knowledge Enhancer (TKE)**: LLMs are trained on text and lack understanding of trajectory patterns, thus we design TKE to **help SLM/LLM understand trajectory data and learn the TDP knowledge**. TKE does not directly apply prompting, LoRA, and so on. Instead, it introduces (a) **trajectory-aware instruction templates** tailored to various TDP tasks, (b) **trajectory offsite-tuning** adapted to SLM and LLM co-training in FL, (c) **LoRA
>   sparse-tuning** guided by layer change ratios, and (d) **bidirectional KD** that jointly aligns SLMs and LLMs across
>   silos. These designs are uniquely required to achieve accurate and generalized TDP reasoning.
> - **Federated Parallel Optimization (FPO)**: Training the above components naively would be **prohibitively slow and communication-intensive**. FPO is our optimization strategy that rethinks the entire training workflow: **split learning** decouples the client and server training, **alternating optimization** freezes the necessary data to reduce communication, and **parallel execution** enables concurrent optimization of TPA, SLM, and LLM objectives, which is key to our training efficiency.
>
> Thus, while our framework **draws inspiration from prior LLM and FL research as any practical system and work must**,
> our components are **problem-specific designs** to a crucial challenge, and together they form a
> **synergistic framework** under the demanding and complex requirements of F-TDP. The resulting system is therefore a
> complex and necessary design rather than a loose and simple combination of existing techniques.
> ```
> W2. The motivation for the strong privacy threat model is kind of weak. The paper lacks a compelling reason for why trajectory data preparation requires such stringent protection against a semi-honest server. It should be more illustrated.
> ```
> We thank the reviewer for raising this important point. We agree that the motivation for our privacy threat model should
> be more explicitly emphasized, and we will revise the paper accordingly. Here we provide additional clarification.
>
> TDP tasks often necessitate considering the data context [1--3], which involves the exchange and sharing of data and
> demands collaborative processing across clients (i.e., cross-client TDP), raising privacy concerns. Such data sharing
> may reveal highly sensitive information, including: home/work locations, fine-grained movement patterns, lifestyle
> routines, and cross-border movements of individuals. As shown in Fig. 1, if the data p is missing, the Fengtai region
> needs to utilize the context of the missing data (i.e., $p$ and $p_2$) for data imputation. However, due to data
> privacy constraints, the Fengtai region cannot access $p_1$ from the Dongcheng region. Consequently, ensuring the
> privacy of trajectory data thus constitutes the challenge the FedTDP framework must address.
>
> In the revised manuscript, we incorporate these clarifications to better motivate the necessity of privacy
> assumptions in our framework.
>
> *[1] Open Anomalous Trajectory Recognition via Probabilistic Metric Learning. IJCAI, 2023.*
>
> *[2] KAMEL: A Scalable BERT-based System for Trajectory Imputation. VLDB, 2023.*
>
> *[3] GraphMM: Graph-based Vehicular Map Matching by Leveraging Trajectory and Road Correlations. TKDE, 2024.*

---

> > ### Author Response · Authors · 2025-11-19
> >
> > ```
> > W3. The claimed "unified" and "generalized" framework relies on manually configured, task-specific prompts and external information. This suggests the generalization is not autonomous but is significantly enabled by human-in-the-loop engineering for each task, limiting the claimed novelty.
> > ```
> >
> > We appreciate the reviewer’s comment, but we respectfully disagree with the concern. We argue that **the use of
> > lightweight prompts does not undermine our claim that FedTDP is a unified and generalized solution for F-TDP**. The key
> > lies in distinguishing between system-level generalization and per-task instruction specification. Our claim of "
> > unified" and "generalized " are based on the following core facts:
> >
> > - **Single Model for 10 Diverse Tasks**: FedTDP eliminates the need for training multiple separate and task-specific
> >   models. All tasks are processed by the **same unified architecture using a single multi-task training objective** (
> >   see Eq. 22 in Appendix C.5). This stands contrast to prior TDP works that require a customized model per task or per
> >   dataset, and it reflects genuine **unification at the model-design level**.
> > - **Strong Cross-Task and Cross-Dataset Generalization**: As shown in Table 2, a model trained on subset tasks within
> >   the GeoLife dataset achieves superior performance across all 10 tasks, including those unseen during training, and 6
> >   different datasets. **This strong generalization is a direct result of the unified knowledge learned through
> >   multi-task learning.**
> >
> > Regarding the prompt, we acknowledge its role but emphasize it is **not a limitation to generalization**. Prompt is the
> > lightweight readable instruction designed to guide SLM/LLM better comprehend data and tasks, which is not a complex
> > and manually crafted algorithm or a trainable model component. While minor textual differences exist across tasks, the
> > majority of the prompt template is **shared, reusable, and standardized**, with only a subset necessitating additional
> > information.
> >
> > The reviewer’s concern seems to equate the existence of input instructions with a lack of autonomy. However, modern
> > LLM-driven systems universally rely on prompts as the mechanism for task communication. Our contribution does not lie in
> > eliminating prompting, but in enabling a **single and unified model backbone** to interpret diverse prompts and perform
> > a wide array of TDP tasks without any architectural modification or parameter re-training. Importantly, **all other
> > modules** that genuinely determine model architecture are entirely **task-agnostic** without manual and task-specific
> > configuration. These components operate identically across all tasks and datasets, deciding the unified and
> > generalized nature of the framework, which is also **acknowledged by Reviewer i8Ui.**
> >
> > In summary, FedTDP is unified because it provides **one coherent and task-agnostic model architecture that generalizes
> > across tasks and datasets**, while prompts serve only as minimal semantic descriptors rather than as a bespoke model
> > engineering.
> >
> > ```
> > Q1. The TPA structure is a lightweight MLP. How does the reconstruction error of the autoencoder impact the accuracy of downstream tasks that require high spatial precision, such as Map Matching?
> > ```
> >
> > We thank the reviewer for this valuable question. TPA is indeed a lightweight MLP, but its design explicitly aims to
> > preserve the spatio-temporal structures that are essential for TDP tasks, including those requiring high spatial
> > precision such as map matching. During training, TPA jointly minimizes reconstruction error and maximizes the TDP
> > performance, ensuring that the learned embeddings retain the geometric and temporal cues necessary for precise
> > trajectory reasoning. As shown in our ablation results (Section 5.2), TPA **does not lead to significant degradation in
> > the accuracy of TDP tasks**, demonstrating that the embeddings are sufficient for LLM to handle TDP tasks effectively.

---

> > > ### Author Response · Authors · 2025-11-19
> > >
> > > ```
> > > Q2. The privacy guarantee assumes the TPA model is private. In the federated setting, how is the secrecy of the aggregated TPA model's architecture and parameters maintained against the semi-honest server?
> > > ```
> > >
> > > We appreciate the reviewer’s question. The privacy of the TPA is rigorously protected against a semi-honest server. To
> > > ensure end-to-end privacy, we deploy two safeguards:
> > >
> > > - **To prevent embedding inversion attacks**, we ensure that the TPA model is kept private and never exposed to the
> > >   server. The server only receives encoded embeddings without access to the embedding model parameters, thereby breaking
> > >   the necessary precondition for embedding inversion. Without knowledge of the embedding model, reconstructing raw
> > >   trajectories from embeddings is infeasible, where a theoretical privacy leakage analysis of the embeddings is provided
> > >   in Appendix C.2.
> > > - **To prevent gradient inversion attacks during TPA model aggregation**, we design a decentralized secret-sharing
> > >   protocol for secure model aggregation. Instead of exchanging raw gradients, the TPA model is split into several
> > >   parameter blocks that are then masked using pairwise secret keys before transmission. Then, each client is responsible
> > >   for aggregating one block without access to other clients' model updates, ensuring that no client can reconstruct
> > >   another client’s data from the gradients. The correctness of this aggregation scheme is formally proven in Appendix
> > >   C.1.
> > >
> > > Overall, our privacy guarantee stems from **isolating the TPA model from the server** and **protecting the TPA model
> > > during the federated optimization process**.
> > >
> > > ```
> > > Q3. (1) How much of the model's generalization to unseen tasks is due to the inherent few-shot ability of the base LLM versus the specific design of the TKE? (2) Could a centrally fine-tuned LLM with good prompts achieve similar results without the federated complexity? These questions could be more emphasized.
> > > ```
> > >
> > > We appreciate the reviewer’s insightful questions and agree that these points are important and deserve further
> > > emphasis. Below, we clarify the distinct roles of the base LLM, the proposed TKE, and the necessity of the federated
> > > setting.
> > >
> > > (1) The base LLM possesses inherent generalization to unseen tasks, and our overall performance experiments confirm that
> > > LLM-based baselines can achieve moderate performance on unseen TDP tasks. However, the performance gains by the base LLM
> > > are limited, and the key improvement arises from the proposed TKE. By guiding the LLM to better understand
> > > trajectory data and introducing the TDP knowledge, TKE substantially amplifies the LLM’s generalization in TDP tasks, improving its performance on both seen and unseen TDP tasks. Moreover, our ablation studies show a clear and
> > > significant performance drop when TKE is removed, indicating that the **LLM’s inherent generalization alone is
> > > insufficient for effective TDP reasoning** and that **TKE plays a crucial role in enabling strong generalization to
> > > TDP tasks**.
> > >
> > > (2) Our overall performance experiments already include LLM-based baselines with evaluated good prompts, which achieve
> > > reasonable but clearly limited performance. This demonstrates that **the prompt alone is insufficient for TDP tasks**.
> > > In contrast, our TKE module helps the LLM to better understand trajectory data and introduces the TDP knowledge,
> > > which is essential for achieving the strong performance observed in experiments. Moreover, due to strict privacy
> > > constraints that **prohibit sharing raw trajectories** across regions in real-world scenarios, federated trajectory
> > > setting is therefore not an optional choice, but a **practical necessity**.
> > >
> > > In the revised manuscript, we expand the discussion to emphasize these questions.
> > >
> > > ---
> > >
> > > Once again, we sincerely appreciate the reviewer’s valuable comments and constructive suggestions. We hope that our
> > > clarifications have adequately addressed the raised concerns, and we would be glad to provide any additional details if
> > > needed. If the reviewer finds the responses satisfactory, we would be grateful for a reconsideration of the overall
> > > score.

---

### Official Review · Reviewer_2n9U · 2025-11-01

**Soundness:** 3
**Presentation:** 3
**Contribution:** 3
**Rating:** 6
**Confidence:** 3

**Summary:**

This paper proposes FedTDP, a unified federated framework to address Trajectory Data Preparation (TDP). The authors identify two major limitations in existing methods: (L1) they require centralized data, which is unrealistic due to privacy laws, leading to "vertically partitioned" data silos (e.g., by region); and (L2) they are task-specific (e.g., one model for imputation, another for map matching) and lack generalization. To solve this, FedTDP proposes a complex server-client architecture with a central LLM and on-device SLMs. Its core innovations are three-fold, designed to solve three new challenges: The framework is evaluated on 10 TDP tasks and 6 real-world datasets, where it reportedly outperforms 13 state-of-the-art baselines.

**Strengths:**

1.	Novel Problem Formulation: The paper clearly defines F-TDP (Federated Trajectory Data Preparation) as a novel problem. It accurately targets two real-world challenges: (L1) vertical data partitioning due to privacy silos and (L2) the need for a general-purpose framework to replace 10+ single-task models.
2.	Strong Experimental Results: This is the paper's greatest strength. As shown in Table 2, FedTDP consistently and significantly outperforms 13 baselines across 10 TDP tasks and 6 datasets.
3.	Excellent System Design and Ablation: The ablation study in Figure 5 is highly effective. It clearly demonstrates the performance.

**Weaknesses:**

1.	Misleading Privacy Guarantees: The paper's central privacy claim is unsound. It rejects Differential Privacy (DP) for degrading utility and claims its TPA + Secret Sharing mechanism provides "formal privacy guarantees". The paper's entire privacy for data-in-transit relies on "computational infeasibility", which is a (weaker) "encryption-like" guarantee, not the "formal" (mathematical, statistical) guarantee that DP provides. This core contradiction undermines the C1 contribution.
2.	Overly Complex System: The FedTDP framework is a massive, complex system integrating numerous distinct technologies (TPA, Secret Sharing, SLMs, LLMs, Offsite-tuning, Sparse-tuning, KD, Split Learning, etc.). This "kitchen sink" approach makes the system extremely difficult to reproduce and analyze.
3.	Questionable Theoretical Analysis: The mathematical derivation for Theorem 2 (LoRA sparse-tuning probability) in Appendix C.4 is overly complex and its correctness is not immediately obvious.

**Questions:**

Q1: Critical Privacy Question: Your privacy analysis relies on the server not being able to reverse the embedding because the encoder is private. The paper's threat model assumes that the server is “semi-honest”, but that the server receives the client's data embedding when performing cross-client tasks. Why is this approach (sending data embeddings) inherently more privacy-secure than “sending gradients” (vulnerable to gradient reversal attacks), which the paper criticizes?
Q2: TKE Complexity: The TKE module is key to the model's high performance but is extremely complex, with four sub-components (prompts, offsite-tuning, sparse-tuning, KD) . Have you conducted an ablation within TKE to identify which of these components are essential? Is it possible the performance gain comes from just one or two (e.g., only offsite-tuning and prompting)?

---

> ### Author Response · Authors · 2025-11-19
>
> We express our gratitude to the reviewer for providing constructive feedback on our paper, and we greatly appreciate the
> acknowledgment of our contributions. Please find below our answers to all the concerns and questions.
> ```
> W1. Misleading Privacy Guarantees: The paper's central privacy claim is unsound. It rejects Differential Privacy (DP) for degrading utility and claims its TPA + Secret Sharing mechanism provides "formal privacy guarantees". The paper's entire privacy for data-in-transit relies on "computational infeasibility", which is a (weaker) "encryption-like" guarantee, not the "formal" (mathematical, statistical) guarantee that DP provides. This core contradiction undermines the C1 contribution.
> ```
>
> We appreciate the reviewer’s careful reading and concern regarding our privacy claims. However, we respectfully disagree
> with the assertion that our privacy guarantee is misleading or unsound. The critique conflates two **distinct privacy
> paradigms**: DP and our approach (i.e., TPA), and applies the evaluation criteria of one to the other. We argue that:
>
> - We do not dispute the security of DP itself, which is indeed well-suited for applications such as **statistical
>   release** or **data synthesis**. However, F-TDP focuses on the **processing of individual trajectory records**, where
>   DP encounters inherent and critical limitations:
>     - When DP-perturbed trajectories are shared as individual records rather than aggregated statistics, it remains
>       possible to **probabilistically infer the original location or time range** within bounds defined by the privacy
>       budget [1-2].
>     - DP enforces privacy at **the cost of utility**, introducing noise that inevitably distorts the delicate
>       spatio-temporal correlations (e.g., speed, direction, turning angles) essential to trajectory data, which is
>       unacceptable for TDP tasks, rendering the resulting data unsuitable for TDP tasks.
> - Our approach is not a competitor to DP but an **alternative and different paradigm**: privacy achieved through secure
>   computation and minimal data exposure. The security of our TPA aggregation protocol is formally proven under the
>   standard semi-honest adversary model, which is analogous to the **formal guarantees provided by secure multi-party
>   computation or homomorphic encryption that rely on computational hardness assumptions**.
>
> Overall, while both DP and TPA aim to preserve privacy, they operate under **distinct assumptions and mechanisms**.
>
> *[1] Geo-Indistinguishability: Differential Privacy for Location-Based Systems. CCS, 2013.*
>
> *[2] Understanding location privacy of the point-of-interest aggregate data via practical attacks and defenses. TDSC,
> 2022.*

---

> ### Author Response · Authors · 2025-11-19
>
> ```
> W2. Overly Complex System: The FedTDP framework is a massive, complex system integrating numerous distinct technologies (TPA, Secret Sharing, SLMs, LLMs, Offsite-tuning, Sparse-tuning, KD, Split Learning, etc.). This "kitchen sink" approach makes the system extremely difficult to reproduce and analyze.
> ```
>
> We appreciate the reviewer’s concern regarding the system complexity. We fully agree that FedTDP is a complex system,
> but we respectfully argue that **this complexity is inherent to the F-TDP problem** itself rather than a result of a
> “kitchen sink” design. Each component in FedTDP are designed and adapted to directly address a concrete and crucial
> challenge arising from the federated trajectory preparation setting.
>
> The F-TDP problem is intrinsically multi-faceted: it simultaneously demands (1) **multi-task generalization**,
> (2) **privacy preservation under vertical data partitioning**, (3) **adaptation of LLMs to trajectory semantics**,
> and (4) **efficient federated optimization**. We argue that a simpler system would inevitably fail to address one or
> more of these requirements. Below, we clarify the necessity and role of each component:
>
> - **LLM/SLM Co-design**: Existing TDP methods are single-task and lack generalization. A unified F-TDP system must
>   support various heterogeneous tasks. This motivates incorporating LLMs, which natively provide **multi-task and
>   cross-domain generalization**. This is not an arbitrary design choice but a requirement of the problem. However, LLMs
>   cannot be deployed and trained on resource-constrained clients. To **enable distributed computation**, we split the
>   workload that introduces a lightweight SLM to handle local TDP tasks on the client, while the server-hosted LLM
>   handles cross-client TDP for trajectories spanning multiple clients. This design **fully utilizes client computation
>   resources** and **reduces server overload**, which is essential for practical FL.
> - **Trajectory Privacy AutoEncoder (TPA)**: Cross-client TDP requires **collaborative data processing**, which
>   necessitates data sharing and raises severe privacy concerns. To **protect trajectory privacy**, we propose TPA that
>   maps raw trajectories to embeddings while preserving the spatio-temporal correlations. To **prevent inversion attacks**, we keep the TPA model private in clients with a lightweight secret sharing scheme for its federated secure
>   aggregation.
> - **Trajectory Knowledge Enhancer (TKE)**: LLMs are trained on text and lack understanding of trajectory patterns, thus
>   we design TKE to **help SLM/LLM understand trajectory data and learn the TDP knowledge**. Its four sub-components are
>   all essential: a) **Trajectory Prompt Engineering** structures the input and integrates essential information,
>   b) **Trajectory Offsite-Tuning** leverages the LLM’s knowledge to boost the SLM’s learning capacity, c) **LoRA
>   Sparse-Tuning** drastically reduces the number of trainable parameters, and
>   d) **Bidirectional Knowledge Distillation** maintains consistent semantics between SLM and LLM. Each subcomponent
>   addresses a distinct limitation of applying LLMs to TDP.
> - **Federated Parallel Optimization (FPO)**: Training the above components naively would be **prohibitively slow and
>   communication-intensive**. FPO is our optimization strategy that rethinks the entire training workflow: **split
>   learning** decouples the client and server training, **alternating optimization** freezes the necessary data to reduce
>   communication, and **parallel execution** enables concurrent optimization of TPA, SLM, and LLM objectives, which is
>   key to our training efficiency.
>
> Regarding reproducibility, we provide a comprehensive open-source implementation at our anonymous repository, which
> includes all code, data, and detailed instructions.
>
> In summary, **FedTDP’s complexity emerges from the complex and realistic constraints of the F-TDP problem**. Each
> component is a necessary response to a crucial challenge, and together they form a synergistic framework, where the
> ablation study demonstrates the effectiveness of each module. Therefore, we believe this design is a strength, not a
> weakness, as it enables a practical and effective solution for a previously unaddressed problem.

---

> ### Author Response · Authors · 2025-11-19
>
> ```
> W3. Questionable Theoretical Analysis: The mathematical derivation for Theorem 2 (LoRA sparse-tuning probability) in Appendix C.4 is overly complex, and its correctness is not immediately obvious.
> ```
>
> We appreciate the reviewer’s attention to the theoretical aspects of our work. We respectfully note that the complexity
> of the proof in Theorem 2 does not arise from unnecessary derivation overhead, but from the intrinsic nature of the
> problem itself. The core of LoRA sparse-tuning selects $M$ layers out of $N$ total layers sequentially, and after each
> selection, the remaining layers must be re-normalized according to their updated probabilities. This creates a dependency
> structure where the selection probability of each layer changes dynamically after every draw. Our derivation in the Appendix
> C.4 explicitly models this process step by step via mathematical induction:
>
> - **Base case**: The probability that layer $L_i$ is chosen first is simply its ratio: $P_1(L_i) = R(L_i)$. For the
>   second draw, if the first selected layer is $L_{j_1}$, the probability that $L_i$ is selected becomes $\frac{R(L_i)
>   }{1 - R(L_{j_1})}$. Summing over all valid choices of $j_1$ yields the closed-form expression for $P_2(L_i)$. A
>   similar renormalization occurs for the third draw, leading naturally to the expression of $P_3(L_i)$.
> - **Recursive step**: Extending this sequential reasoning, the probability that $L_i$ is selected at the $k$-th step
>   requires summing over all $k-1$ length valid selection histories, each with appropriately normalized denominators.
>   This yields the general expression for $P_k(L_i)$, and consequently the final selection probability $P(L_i, M) =
>   \sum_{k=1}^{M} P_k(L_i)$.
> - **Correctness verification**: For any selection step $k$, the sum of the conditional probabilities over all remaining
>   layers is exactly 1, because the numerator $\sum_{i \notin \\{j_1,\dots,j_{k-1}\\}} R(L_i) = 1 - \sum_{t=1}^{k-1} R(L_
>   {j_t})$ cancels the same expression in the denominator. Summing this property over all $k$ leads to $\sum_{i=1}^{N} P(
>   L_i, M) = M$, as expected.
>
> In the revised manuscript, we include a more detailed analysis and correctness verification of Theorem 2 to enhance
> clarity.
>
> ```
> Q1. Critical Privacy Question: Your privacy analysis relies on the server not being able to reverse the embedding because the encoder is private. The paper's threat model assumes that the server is “semi-honest”, but that the server receives the client's data embedding when performing cross-client tasks. Why is this approach (sending data embeddings) inherently more privacy-secure than “sending gradients” (vulnerable to gradient reversal attacks), which the paper criticizes?
> ```
>
> We appreciate the reviewer’s valuable question. We clarify that our approach **does not claim that “sending embeddings”
> is inherently more secure than “sending gradients”**. In fact, as we explicitly state in Section 4.1, both raw
> embeddings and raw gradients are vulnerable to privacy attacks through embedding inversion and gradient inversion,
> respectively.
>
> Our key insight is not about the type of data transmitted, but about how we protect it. TPA is a learning-based
> transformation that maps raw trajectory to embeddings, which is designed to protect trajectory privacy while preserving
> spatio-temporal correlations. To ensure end-to-end privacy, we deploy two safeguards:
>
> - **To prevent embedding inversion attacks**, we ensure that the **TPA model is kept private and never exposed to the
>   server**. The server only receives encoded embeddings without access to the embedding model parameters, thereby
>   breaking the necessary precondition for embedding inversion. Without knowledge of the embedding model, reconstructing
>   raw trajectories from embeddings is infeasible, where a theoretical privacy leakage analysis of the embeddings is
>   provided in Appendix C.2.
> - **To prevent gradient inversion attacks during TPA model training**, we design a decentralized secret-sharing protocol
>   for secure model aggregation. Instead of exchanging raw gradients, the TPA model is split into several parameter blocks
>   that are then masked using pairwise secret keys before transmission. Then, each client is responsible for aggregating
>   one block without access to other clients' model updates, which ensures that no client can reconstruct another client’s
>   data from gradients. The correctness of this aggregation scheme is formally proven in Appendix C.1.
>
> Overall, our privacy guarantee stems from **isolating the TPA model from the server** and **protecting the TPA model**
> during federated optimization process, not from assuming that embeddings themselves are unrecoverable.

---

> > ### Author Response · Authors · 2025-11-19
> >
> > ```
> > Q2: TKE Complexity: The TKE module is key to the model's high performance but is extremely complex, with four sub-components (prompts, offsite-tuning, sparse-tuning, KD). Have you conducted an ablation within TKE to identify which of these components are essential? Is it possible that the performance gain comes from just one or two (e.g., only offsite-tuning and prompting)?
> > ```
> >
> > We thank the reviewer for this insightful question. While we agree that understanding the contribution of each
> > sub-component is important, we respectfully argue that the four components of TKE are designed to be **complementary
> > rather than interchangeable**. As such, their effects are not additive or isolated, and removing any single component
> > would not provide meaningful attribution. Specifically,
> >
> > - **Trajectory prompt engineering** provides the LLM/SLM with structured input semantics and task intent, serving as the
> >   foundation that enables all downstream components to operate effectively.
> > - **Trajectory offsite-tuning** transfers the LLM’s strong reasoning and generalization capabilities to the lightweight
> >   SLM equips the client model with general knowledge that cannot be learned from scratch.
> > - **LoRA sparse-tuning** ensures that the additional parameters introduced by TKE are efficient and scalable under
> >   federated constraints.
> > - **Bidirectional knowledge distillation** aligns the representations learned by the SLM and LLM so that both sides
> >   operate coherently across heterogeneous clients.
> >
> > These four elements serve distinct but mutually reinforcing roles, and the strong generalization of FedTDP arises from
> > the synergy among these components rather than any single one.
> >
> > In summary, TKE is intentionally constructed as a mechanism where the four sub-components work together to address
> > different challenges of trajectory reasoning under federated constraints. Its **effectiveness stems from this synergy**,
> > and isolating individual pieces would not yield meaningful insights into its design or capabilities.
> >
> > ---
> >
> > We sincerely appreciate the reviewer’s positive assessment and evaluation. Thank you for recognizing the strengths of
> > our work. If there are any remaining questions or aspects that would benefit from further clarification, we would be
> > very happy to provide additional details.

---

### Author Response · Authors · 2025-11-30
**Global Summary**

We thank the AC and reviewers for their time and effort in providing thorough
evaluations, constructive feedback, and meaningful discussions throughout the review process. Across the
reviews, we observed a clear recognition of the core **strengths and contributions** of our work, including Novel
Problem Formulation, Unified Framework with Novel contributions,Excellent System Design and Ablation,
Rigorous Theoretical Privacy Analysis,Comprehensive Empirical Validation, and Robust Generalization and
Practical Efficiency.

During the rebuttal period, we provided detailed clarifications addressing all **technical concerns** raised by the
reviewers, including privacy guarantees, the necessity of the system components, generalization sources, theoretical
correctness, and scalability considerations. These explanations were met with positive responses: Reviewers **i8Ui** and
**Lr1C** explicitly stated that all their concerns **were fully resolved**. Reviewer **Lr1C** further noted that the
detailed responses **supported a higher rating**.

Overall, the reviewers’ comments and discussions have contributed meaningfully to improving the clarity, rigor, and
completeness of the manuscript. We are sincerely grateful for their constructive insights, which helped refine the
presentation and sharpen the technical narrative of FedTDP. We hope that the revisions and responses clearly demonstrate
the novelty and practical significance of our proposed framework.

---

For clarity, we summarize below how the reviewers’ concerns were resolved and how the rebuttal contributed to
strengthening the submission.

---

> ### Author Response · Authors · 2025-11-30
> **Reviewer 2n9U (Rating: 6, Confidence: 3)**
>
> Reviewer 2n9U provided an extensive and balanced evaluation, acknowledging several major **strengths** of our work,
> including (1) the novelty and clarity of defining Federated Trajectory Data Preparation (F-TDP) as a new problem, (2)
> the strong and consistent empirical improvements across 10 TDP tasks and 6 real-world datasets, and (3) the
> effectiveness of our system design and ablation studies.
>
> The reviewer also raised three main **concerns**: privacy guarantees, system complexity, and the correctness/clarity of
> Theorem 2, as well as two technical questions. We provided detailed clarifications to address each point:
>
> - **Privacy guarantees (W1 & Q1)**: We explained that the reviewer’s critique conflates two fundamentally different
>   privacy paradigms (differential privacy vs. secure-computation based privacy). We clarified the limitations of DP for
>   per-record trajectory processing and emphasized that our TPA module follows the formal guarantees widely used in
>   secure multi-party computation. We further clarified that we do not assume that embeddings are irreversible.
>   Instead, our privacy guarantee relies on (1) keeping the TPA model private blocking embedding inversion and (2) secure
>   decentralized aggregation blocking gradient inversion. This resolved the core misunderstanding. Here, we would like to
>   emphasize that black-box inversion requires either access to input-embedding pairs or query access to the encoder,
>   neither of which is permitted in our threat model.
> - **System complexity (W2)**: We clarified that the system complexity arises naturally from the inherent requirements of
>   the F-TDP problem: multi-task generalization, vertical partitioning privacy, trajectory-LLM adaptation, and efficient
>   federated optimization, where each component addresses a distinct and necessary challenge, and a simpler architecture
>   would fail to meet these requirements.
> - **Correctness & clarity of theorem 2 (W3)**: We provided a clearer inductive derivation of Theorem 2 with more
>   details, and added a correctness check demonstrating that the total selection probability sums to the expected number
>   of selected layers. This directly addressed the reviewer’s concern regarding correctness and clarity.
> - **TKE component roles (Q2)**: We clarified that the four TKE submodules are complementary rather than additive or
>   isolated, where each addresses a different bottleneck in TDP-oriented LLM adaptation. Ablating individual submodules
>   would not yield meaningful insights.
>
> Overall, Reviewer 2n9U’s concerns were fully resolved through our rebuttal, and their review acknowledged the strong
> novelty, empirical results, and system design of our work.

---

> > ### Author Response · Authors · 2025-11-30
> > **Reviewer Dh2g (Rating: 4, Confidence: 3)**
> >
> > Reviewer Dh2g provided a balanced and constructive review, acknowledging several **strengths** of our work: (1) the
> > clear identification of the F-TDP problem, (2) the comprehensive experiments across 10 tasks and 6 real-world datasets, and
> > (3) the overall clarity and structure of the paper.
> >
> > The reviewer also raised **concerns** regarding (1) whether our work is a patchwork of existing techniques, (2) the
> > motivation of the privacy threat model, (3) the unified/generalized claim that relies on manually configured prompts,
> > and (iv) several technical questions. We provided detailed responses that addressed each concern:
> >
> > - **On the alleged “patchwork” design (W1)**: We clarified that although our framework draws on prior ideas (as any
> >   practical system does), each module in FedTDP: TPA, TKE, and FPO, is not used off-the-shelf but is purpose-designed to
> >   address the unique challenges of F-TDP. The reviewer’s concern stemmed from interpreting these components as generic.
> >   Reviewer i8Ui explicitly acknowledged the novelty and necessity of these designs as part of the contribution.
> > - **On the motivation for the privacy threat model (W2)**: We provided concrete and illustrative explanations showing
> >   why trajectory data carries extremely sensitive information, and clarified the necessity of protecting cross-client
> >   contextual information during collaborative TDP. These details have been added to the revised manuscript.
> > - **On the use of prompts and generalization (W3)**: We emphasized the distinction between system-level unification (a
> >   single model architecture and training pipeline solving all 10 TDP tasks) and lightweight instruction specification.
> >   Prompt acts only as minimal task descriptors, not manually crafted algorithm or a trainable model component, while all
> >   other architectural components are task-agnostic and shared across tasks. We demonstrated strong cross-task and
> >   cross-dataset generalization without retraining. This directly addresses the reviewer’s concern about
> >   human-in-the-loop engineering.
> > - **On the impact of TPA in TDP accuracy (Q1)**: We clarified that TPA preserves essential spatio-temporal information,
> >   and ablations confirm that downstream accuracy does not suffer when using TPA embeddings.
> > - **On TPA secrecy (Q2)**: We explained that TPA parameters never leave the client and are protected during
> >   federated aggregation through decentralized secret sharing. This prevents both embedding inversion and gradient
> >   inversion under the semi-honest threat model.
> > - **On generalization vs. TKE vs. centralized prompts tuning (Q3)**: We showed that while base LLMs have moderate
> >   few-shot ability, the majority of performance gains stem from trajectory-aware TKE, as evidenced by the sharp drop
> >   when TKE is removed in the ablation. We further highlighted that LLM-based baselines with evaluated good prompts have
> >   limited performance as the prompt alone is insufficient for TDP tasks. In addition, we explained the federated setting
> >   necessary rather than optional.
> >
> > Overall, the reviewer’s concerns could be thoroughly addressed through our rebuttal, and we believe the clarifications
> > would strengthen the reviewer’s understanding of the novelty, necessity, and coherence of FedTDP.

---

> > > ### Author Response · Authors · 2025-11-30
> > > **Reviewer i8Ui (Rating: 8, Confidence: 5)**
> > >
> > > Reviewer i8Ui provided a highly **positive and supportive** evaluation, recognizing both the novelty and the technical
> > > depth of our work: (1) the well-motivated formulation of F-TDP, (2) the strong generalization ability of the unified
> > > LLM-based framework across 10 tasks and 6 datasets, (3) novel technical contributions with purpose-built modules, and
> > > (4) the rigorous theoretical privacy analysis.
> > >
> > > The reviewer also provided several constructive **suggestions** aimed at further improving clarity and completeness. We
> > > addressed these points in detail as follows:
> > >
> > > - **Inclusion of key experiments (W1)**: The reviewer suggested moving important studies from the appendix to the main
> > >   paper. We fully agreed and have integrated the results of model efficiency and generalization into Section 5 (
> > >   Experiment) of the revised manuscript to improve readability and empirical clarity.
> > > - **Clarify the caption of Figure 10 (W2)**: As recommended, we revised the figure caption of model generalization to
> > >   “Generalization Study” to better reflect its content and avoid ambiguity.
> > > - **More in-depth discussion of limitations (W3)**: We expanded Appendix E with a thorough and structured discussion of
> > >   FedTDP’s limitations. This includes deeper analysis of: (a) the boundary of trajectory-level preprocessing vs.
> > >   semantic-level reasoning and (b) interpretability challenges of LLM-enhanced components. These additions significantly
> > >   strengthen the completeness and transparency of the paper.
> > >
> > > We deeply appreciate the reviewer’s final acknowledgment that our rebuttal **addressed all concerns and significantly
> > > strengthened the paper**, which further confirms that the reviewer’s questions and suggestions were satisfactorily
> > > resolved.
> > >
> > > Overall, Reviewer i8Ui’s evaluation provides strong support for the contribution, clarity, and rigor of our work, and
> > > their constructive feedback helped further polish the manuscript.

---

> > > > ### Author Response · Authors · 2025-11-30
> > > > **Reviewer Lr1C (Rating: 4 $\to$ Improved, Confidence: 5)**
> > > >
> > > > Reviewer Lr1C provided a detailed and critical evaluation, while acknowledging several **strengths** of our work: (1) New Problem Formulation, (2) Privacy-Preserving Design, and (3) Strong Generalization & Efficiency.
> > > >
> > > > The reviewer also raised several **concerns** regarding the paper organization, rigor of the privacy analysis, system complexity, and evaluation design. We addressed these points in our rebuttal:
> > > > - **Conclusion missing in the main text (W1)**: We removed the conclusion content to the main paper in the revised version.
> > > > - **Formal privacy guarantees (W2 & Q1)**: We clarified that our privacy guarantee is grounded in formally proven under the semi-honest adversary model. Specifically, (a) embedding inversion is infeasible because the TPA model never leaves the client and is never queryable, and (b) gradient inversion is blocked by our decentralized secret-sharing aggregation scheme.
> > > > - **Generalization claims and confounding factors (W3 & Q7)**: We explained that generalization is demonstrated under 10 diverse TDP tasks and across 6 datasets covering different cities and mobility patterns. The observed gains cannot be explained by dataset similarity alone and that ablation studies disentangle the roles of pre-training, TKE, and federated optimization, supporting genuine cross-task and -domain generalization.
> > > > - **System complexity (W4)**: We explained that the complexity of FedTDP arises from the realistic requirements of the F-TDP problem, where each module is specially designed to address the real-world challenges of F-TDP. Ablation results support the necessity and coherence of the overall design.
> > > > - **Vertical partition and boundary trajectories (Q2)**: We explained that local and cross-client TDP tasks are explicitly distinguished. Boundary trajectories are securely processed by encoding sub-trajectories locally and aligning them at the server using privacy-preserving embeddings.
> > > > - **Unclear contribution of TKE and FPO (Q3)**: We clarified that the ablation isolates the roles of TKE and FPO by removing them independently, demonstrating clear non-overlapping effects that TKE governs semantic generalization and reasoning ability, while FPO governs training efficiency.
> > > > - **Scalability and communication cost (Q4)**: We emphasized that the communication overhead (~5 GB in total) is well within the capacity of modern FL systems, whose bandwidth commonly supports tens of gigabytes of transfer. Besides, FPO reduces runtime by over one order of magnitude, and the ablation shows that TPA’s additional overhead is small relative to the overall pipeline.
> > > > - **Explainability of LLM adaptation (Q5)**: We clarified that our work already provides initial visualization of learned trajectory-aware representations in Appendix C.6. Besides, we explicitly noted in Appendix E that deeper interpretability of LLM-enhanced components remains an inherent limitation of current LLM-based systems.
> > > > - **Inconsistent evaluation metrics (Q6)**: We clarified that the use of metrics across tasks is standard and reflects the established practice in TDP research, following well-accepted protocols in previous TDP work.
> > > > - **Arbitrary LoRA sparsity choice (Q8)**: We clarified that the parameter sensitivity analysis provides a principled basis for LoRA sparsity choice.
> > > > - **Sparse-sampled trajectories (Q9)**: We noted that evaluated datasets naturally contain sparse and irregular sampling patterns, and FedTDP achieves consistently strong performance. This demonstrates robustness to sparsity levels commonly in real-world trajectories.
> > > >
> > > > In the follow-up discussion, the reviewer also raised two additional concerns:
> > > > - Regarding **black-box embedding inversion and the robustness of secure aggregation**, we clarified that black-box inversion requires either input-embedding pairs or query access to the encoder, neither of which is permitted in our threat model. We also clarified that our decentralized block-wise aggregation remains correct and private even under client dropouts, as each block can be independently aggregated or reassigned.
> > > > - Regarding **system overhead and real-world feasibility**, we clarified that FedTDP’s runtime and communication costs have been evaluated in the efficiency study: all modules are lightweight compared to the overhead introduced by LLM/SLM training, and FPO provides over an order-of-magnitude speedup compared with LLM-based baselines. We also emphasized that training separate models for single-task methods would be significantly more expensive. In addition, Appendix C.5 shows that the complexity is dominated by the underlying LLM/SLM backbone. Overall, the empirical and theoretical results demonstrate that FedTDP is computationally practical for real-world federated deployment.
> > > >
> > > > After reading our rebuttal and clarifications, Reviewer Lr1C explicitly stated that **all concerns were fully resolved
> > > > and that they would improve their rating**.

---

> ### Author Response · Authors · 2025-11-30
> **Summary of Revisions**
>
> In the revised manuscript, we have incorporated all requested clarifications, enhancements, and structural improvements to strengthen the technical rigor and the overall presentation of the work. All modifications are highlighted in **blue** in the revised PDF for ease of review. The major revisions include:
>
> - **In Section 1 (Introduction, page 2)**, we expanded and clarified **privacy challenges (C1)** to
>   better motivate the necessity of privacy assumptions in our framework.
>
>   ——*Specifically, we explained why TDP tasks inherently require cross-client contextual information and how such information exchange can expose sensitive mobility
>   patterns. We also added a concrete cross-region imputation example to illustrate why the client cannot access context
>   from other clients under realistic privacy constraints.*
> - **In Sections 5.2 and 5.3 (Experiment, pages 8-9)**, we expanded the analysis of **overall performance** and
>   **ablation study** to emphasize the source of FedTDP’s generalization ability and the contribution of the TKE module.
>
>   ——*The revised text now (1) highlights that FedTDP achieves substantially larger gains by specific design of the TKE,
>   while LLM-based baselines exhibit limited performance base the inherent few-shot ability of the base LLM, (2) shows
>   through ablation that removing TKE causes a large performance drop, demonstrating that TKE is the main source of
>   generalization across unseen tasks and domains rather than the base LLM, and (3) underscores that simple
>   prompt-based tuning of LLMs cannot match the performance achieved by FedTDP’s trajectory-aware knowledge enhancement
>   and federated learning design.*
> - **In Section 5.4 (Experiment, page 10)**, we moved the **model generalization study** and **efficiency study** from
>   the appendix into the main paper to improve the readability and completeness of the empirical evaluation. We also
>   revised the corresponding figure caption in the generalization study to ensure consistency and clarity.
> - **In Section 6 (Conclusion, page 10)**, we moved the content of conclusion from appendix to the main text improve the
>   paper's completeness.
> - **In Appendix C.2 (Theoretical Privacy Analysis, page 20)**, we added a more rigorous and explicit theoretical
>   treatment of the privacy guarantees in FedTDP.
>
>   ——*The revised text now (1) formally explains how isolating the TPA
>   encoder from the server prevents embedding inversion attacks by removing the prerequisites, and (2) provides a
>   clearer description of the decentralized secret-sharing aggregation protocol, including how masking, block-wise
>   aggregation, and the correctness guarantee jointly defend against gradient inversion attacks.*
> - **In Appendix C.4 (Proof of Theorem 2, pages 21-23)**, we substantially expanded the theoretical explanation of
>   Theorem 2 by providing a complete, step-by-step derivation of the layer-selection probability used in LoRA
>   sparse-tuning.
>
>   ——*The revised text now clearly explains how weighted sampling without replacement operates across
>   multiple selection rounds, why the probability must be renormalized after each draw, and how the final selection
>   probability is obtained by aggregating over all possible draw positions. Additionally, we added a dedicated
>   correctness verification section to confirm that the derived probabilities satisfy the expected selection behavior.*
> - **In Appendix E (Limitations, pages 28-29)**, we expanded the discussion of the framework’s limitations to provide a
>   deeper and more transparent analysis of the boundaries of FedTDP.
>
>   ——*The revised text now (1) clarifies that FedTDP is
>   designed as a trajectory preparation framework rather than an end-to-end semantic analysis engine, explaining why
>   tasks such as clustering, pattern mining, or long-term forecasting may require fundamentally different model
>   capacities, and (2) adds a detailed discussion of interpretability challenges arising from the use of LLMs/SLMs as
>   black-box reasoning modules, including implications for trust, debugging, and potential bias. We also outline
>   promising future directions such as integrating explanation mechanisms and deeper semantic modeling.*
>
> Taken together, these revisions substantially improve the clarity, completeness, and robustness of the manuscript. Each
> change directly addresses reviewer comments and further reinforces the motivation, technical soundness, and empirical
> validity of FedTDP. We hope the revised paper now fully meets the expectations of the reviewers and the program
> committee.

---

### Meta-Review · Area_Chair_3pih · 2026-01-06

**Summary:**

This paper proposed a new Federated Trajectory Data Preparation framework named FedTDP, which can be utilized in the scenario where trajectories are vertically partitioned across regions and cannot be directly shared. Four reviewers provided detailed comments on the paper, and their scores are 6,4,8,4. Reviewer Lr1C who gave the score 4 is satisfied with the rebuttal and promise to raise the score. Generally, the reviewers agree the paper studied an interesting and practically important problem, and has clear technique novelty. But they also have a lot of concerns on the privacy guarantee, the over claimed contribution, and the heavy model. To sum, I recommend to accept the paper.

**Reviewer Concerns:**

Addressed:
Reviewer 2n9U: Correctness & clarity of theorem 2, TKE component roles
Reviewer Dh2g: the motivation for the privacy threat model, the impact of TPA in TDP accuracy
Reviewer i8Ui: all are addressed

Outstanding:
Reviewer 2n9U: The model is too heavy
Reviewer Dh2g: The over claimed "unified" and "generalized" framework

Reviewer Lr1C's concerns are well addressed and she/he would like to raise the score.

**Reviewer Scores:**

Reviewer Lr1C will raise the score as she/he said in the discussion. Reviewer 2n9U and Dh2g are both likely to increase their scores.

---

### Decision · Program_Chairs · 2026-01-26

Accept (Poster)